# The Donnan potential revealed

Pinar Aydogan Gokturk ®[1], Rahul Sujanani[2], Jin Qian[1,3], Ye Wang ®[1,3], Lynn E. Katz[4], Benny D. Freeman[2] & Ethan J. Crumlin ®[1,3] ✉

Selective transport of solutes across a membrane is critical for many biological, water treatment and energy conversion and storage systems. When a charged membrane is equilibrated with an electrolyte, an unequal distribution of ions arises between phases, generating the so-called Donnan electrical potential at the solution/membrane interface. The Donnan potential results in the partial exclusion of co-ion, providing the basis of permselectivity. Although there are well-established ways to indirectly estimate the Donnan potential, it has been widely reported that it cannot be measured directly. Here we report the first direct measurement of the Donnan potential of an ion exchange membrane equilibrated with salt solutions. Our results highlight the dependence of the Donnan potential on external salt concentration and counter-ion valence, and show a reasonable agreement with current theoretical models of IEMs, which incorporate ion activity coefficients. By directly measuring the Donnan potential, we eliminate ambiguities that arise from limitations inherent in current models.

The electrochemical potential distribution at a membrane/solution interface profoundly influences the transport and selectivity of solutes across semi-permeable membranes and plays an important role in many electrochemical[1], biological[2,3], and colloidal systems[4]. For naturally occurring systems, such as biological cells, well-regulated control of ion transport is achieved through highly specialized ion channels capable of displaying extraordinary degrees of selectivity[3]. These channels are responsible for the initiation and continuation of electrical signals in the nervous system[5], the timely delivery of $Ca^{2+}$ ions that initiate a contraction in muscle cells[6], the regulation of volume increase/decrease in response to transient changes in the cell environment[7], and pH sensing and survival of bacteria under acid stress[8,9]. Similarly, various technologies for water purification and energy storage and conversion rely on ion exchange membranes (IEMs) to selectively control ion transport rates. IEMs contain ionizable groups, bound to the membrane, that allow permeation of oppositely charged ions (counter-ions), while largely rejecting similarly charged ions (co-ions)[10–14]. In 1911, the permselective nature of IEMs was postulated for the first time by Donnan, using thermodynamic equilibrium and electrostatic considerations[15]. Due to the presence of fixed ions and the condition of electroneutrality, an unequal distribution of ions

exists in a charged polymer, generating an electrical potential, the Donnan potential, at the membrane/solution interface. The Donnan potential acts to exclude co-ions from entering the membrane[16], where larger absolute Donnan potential values lead to stronger co-ion exclusion from the membrane[17]. To date, Donnan's theory has been the accepted framework for describing ion sorption phenomena in IEMs, although disagreements exist between experimentally determined ion sorption values and values derived from the original theory[18–20]. Despite the long history and widespread use of IEMs, direct measurement of the Donnan potential and experimental verification of Donnan's framework has been impossible due to the lack of appropriate interfacial experimental techniques, which has held back the development of novel membranes that are more selective for specific ions (e.g., Li for resource recovery, Ra for hazardous waste management, As for water treatment).

We therefore sought to develop an experimental methodology for directly measuring the Donnan potential in a commercial ion-exchange membrane equilibrated with monovalent and divalent aqueous salt solutions (i.e., NaCl and $MgCl_2$) under equilibrium conditions. Our measurements are based on the binding energy shift in membrane related core levels detected by tender ambient pressure X-ray

[1]Advanced Light Source, Lawrence Berkeley National Laboratory, Berkeley, CA 94720, USA. [2]McKetta Department of Chemical Engineering, The University of Texas at Austin, Austin, TX 78712, USA. [3]Chemical Sciences Division, Lawrence Berkeley National Laboratory, Berkeley, CA 94720, USA. [4]Department of Civil, Architectural, and Environmental Engineering, The University of Texas at Austin, Austin, TX 78712, USA. ✉e-mail: ejcrumlin@lbl.gov

photoelectron spectroscopy (tender-APXPS). This work opens a new avenue to broadly rationalize ion transport and selectivity in membranes by directly probing the Donnan potential developed at the solution/membrane interface.

## Results and discussion

### Probing the Donnan potential with tender APXPS

In this study, aqueous salt solutions with concentrations ranging from 0.001 to 1 M were equilibrated with CR-61, a commercial poly(p-styrene sulfonate-co-divinylbenzene) cation exchange membrane (i.e., a membrane bearing negatively charged sulfonate groups). Details of the membrane properties are available in "Methods" and Supplementary Table 1. Prior to tender-APXPS measurements, we used the "dip and pull" method[21–23] to form a thin solution layer on the membrane surface. The solution layer was estimated to be about 17-21 nm thick (details regarding this calculation are given in Supplementary Note 1). Tender-APXPS measurements at the solution/IEM interface were performed using a photon energy of 4.0 keV (Fig. 1a)[21,23,24]. Representative survey spectra from a CR-61 membrane/salt solution interface are provided in Supplementary Fig. 1. When two phases (e.g., solution and IEM) containing mobile ionic species are brought into contact, ions diffuse in response to electrochemical potential gradients. However, the fixed charge groups in the membrane ($-SO_3^-$ groups in CR-61) cannot do so because they are covalently bound to the membrane network. Electroneutrality requires the fixed charge groups of the membrane to be balanced with an equivalent number of counter-ions (oppositely charged ions that balance the charge of the $SO_3^-$ groups). Co-ions (ions having the same charge as the fixed charge groups) sorbing into the membrane must also be accompanied by an equivalent number of counter-ions to balance the co-ion charge. At equilibrium, therefore, an unequal distribution of ions between the membrane and solution arises (Fig. 1b) and creates an electrical potential, the Donnan potential, at the membrane/solution

interface[16,17]. The Donnan potential restricts co-ion sorption into the membrane and counter-ion desorption into the solution (i.e., Donnan exclusion). Its magnitude depends on the difference in thermodynamic activity between the counter-ions in the solution and membrane phases (see "Methods"). Thus, when the external solution concentration is low relative to the fixed charge concentration of the membrane, the absolute value of the Donnan potential is high (Fig. 1c). A facile but relatively under-utilized feature of photoelectron spectroscopy is its ability to detect local potentials[22,24,25], and we used this capability to directly measure the electric potential drop at a membrane/solution interface. At equilibrium, the binding energies (BE) of the membrane-related core levels shift to lower binding energies, due to the decreasing electric potential in the membrane phase. This shift in binding energy is directly related to the Donnan potential ($\Psi_D$) by $\Delta BE = \Delta\Psi_D$ eV (Fig. 1d, e). (The energy level diagram of the solid/liquid/gas system appears in Supplementary Fig. 2).

The representative O 1s and S 1s core level spectra collected at the membrane/solution interface under equilibrium conditions are shown in Fig. 2a, b, respectively, as a function of external salt concentration. The O 1s spectra (Fig. 2a) were deconvoluted into three different chemistries attributed to gas phase water (GPW), liquid phase water (LPW), and membrane $SO_3^-$ groups, while the S 1s spectra (Fig. 2b) exhibit a single chemistry originating from membrane $SO_3^-$ groups. Details regarding peak fitting parameters and constraints can be found in Supplementary Table 5. At thermodynamic equilibrium, the electric potential of the bulk electrolyte should be zero and independent of solution concentration. Due to the unequal distribution of counter- and co-ions sorbed into the membrane, however, the potential of the membrane changes relative to that of the bulk electrolyte. For this reason, we aligned the BE of electrolyte related LPW peaks in the O 1s region and followed the BE differences in the membrane related S 1s peak originating from the sulfonate groups of the membrane. We note that, depending on the electrolyte concentration, the thickness of the

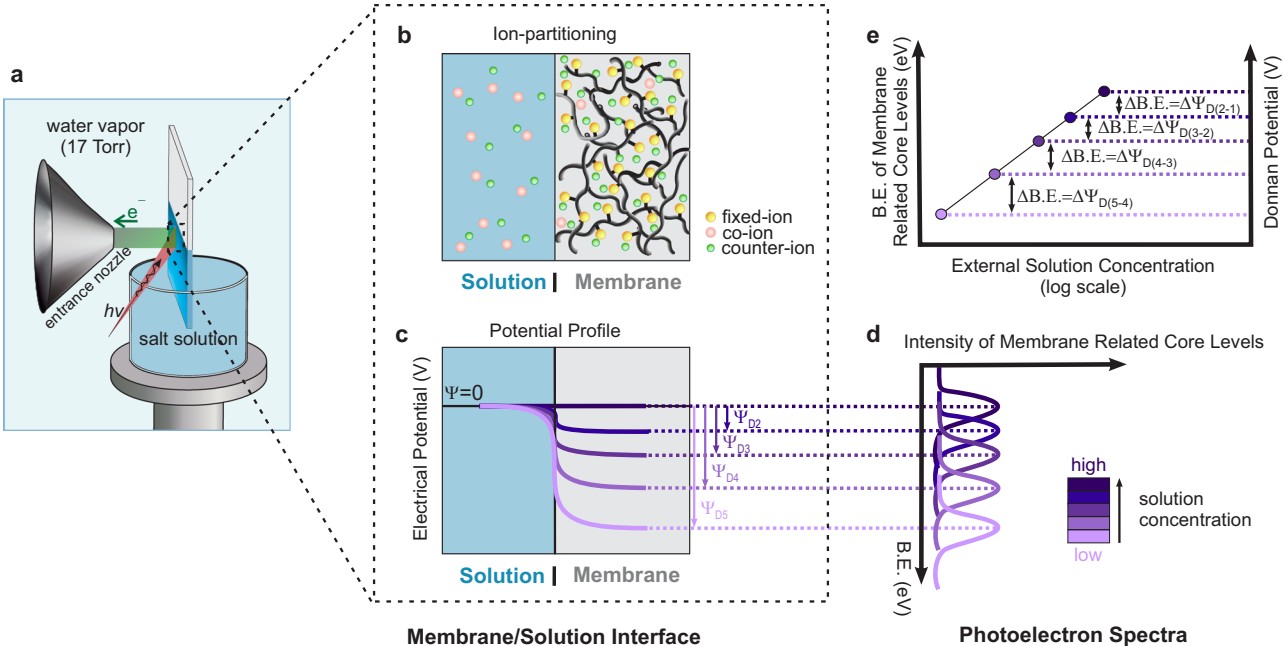

**Fig. 1 | Overview of direct Donnan potential measurement using tender APXPS. a** A schematic of the experimental setup during data acquisition from the CR-61 membrane equilibrated in aqueous salt solution. **b** Equilibrium ion-partitioning between a membrane in contact with a monovalent electrolyte solution. At equilibrium, electroneutrality requires an equivalent number of counter-ions (which, for a monovalent system, is equal to the fixed charge concentration) to balance the fixed charge groups. The co-ions sorbing into the membrane are accompanied by an equivalent number of counter-ions, leading to significantly higher concentrations of counter-ions than co-ions in the membrane. **c** Illustration of the potential drop at the membrane/solution interface. **d** Corresponding binding energy (BE) shifts in the membrane related core levels. **e** Relationship between the measured BE of membrane related core levels and the Donnan potential as a function of external solution concentration.

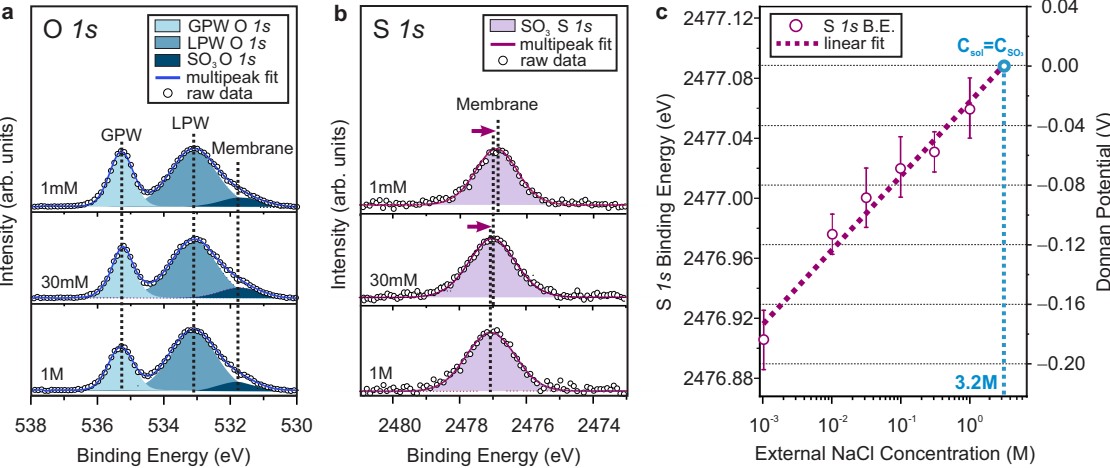

**Fig. 2 | Experimental measurements of the Donnan potential.** Representative **a** O *1s* and **b** S *1s* core level spectra collected from CR-61 membranes equilibrated with different concentrations of NaCl solutions (GPW: gas phase water, LPW: liquid phase water). The binding energy is calibrated using bulk liquid phase water. The effects of the EDL on the binding energy position of the LPW peak are considered, and corrections are made during the calibration process. (Details of the energy calibration process are set forth in Supplementary Note 4) Circles represent raw experimental data, and lines indicate the sum of fits. Representative spectra not provided here for three other concentrations of NaCl solutions are presented in Supplementary Fig. 6 for visual inspection. **c** Averaged S *1s* binding energy and the corresponding Donnan potential values as a function of the external solution concentration. Note that the S *1s* binding energy shift versus Donnan potential shows a linear dependence of ~1 eV/V. Error bars that represent the experimental uncertainty in S *1s* binding energy were determined from the standard deviation of repeated measurements of a minimum of five different locations in each case. The binding energy values of fitted core level components for each individual analysis position are provided in Supplementary Table 6. The dashed line represents the best fit to a linear dependence. The fitted line was extrapolated to the equivalent concentration of counter-ions (3.2 M), where the Donnan potential approaches 0 V[17].

Electrical Double Layer (EDL) formed on the solution side of the interface changes, and when the EDL thickness becomes comparable to the experimental probing depth, electrolyte related core level peaks undergo BE shifts and asymmetric spectral broadening (Supplementary Fig. 3)[24]. To account for this effect, especially evident at lower electrolyte concentration, we simulated the potential distribution at a charged membrane/solution interface in which the membrane fixed charges are uniformly distributed throughout the membrane by solving the nonlinear Poisson-Boltzmann equation for regions both inside and outside of the membrane (Methods and Supplementary Note 2 provides details for potential distribution simulations). Corresponding BE shifts in LPW were estimated by simulating XP spectra from the potential distributions within the EDL and then corrected during the energy calibration process (Supplementary Notes 3 and 4). Figure 2c shows the binding energy shift in the membrane-related S *1s* core level as a function of external NaCl concentration. To convert the measured binding energy shift of the membrane S *1s* core level to the corresponding Donnan potentials at specific external solution concentrations, we extrapolated the linear fit of the experimental data to 3.2 M, where the concentration of the external solution is approximately equal to the concentration of fixed ions in the membrane. The specific binding energy at 3.2 M is aligned to the 0 V Donnan potential because the Donnan potential should be nearly zero when the counter-ion activity in the membrane is equal to the counter-ion activity in the solution, which we approximate as 3.2 M for NaCl[26]. Details of Donnan potential calculations and the relevant equations appear in the "Methods".

We note that the Donnan potential of the CR-61 membrane equilibrated with an aqueous salt solution can also be extracted from the binding energy shifts of other membrane related core levels (i.e., O *1s*-membrane and C *1s*). A discussion about other core levels is presented in Supplementary Note 5 and 6.

## Effect of counter-ion valence
We also assessed the influence of counter-ion valence (i.e., monovalent versus divalent counter-ions) on the Donnan potential of CR-61 membranes equilibrated with various concentrations of NaCl and

MgCl₂ solutions (Fig. 3). Energy calibrated O *1s* and S *1s* core level spectra collected from CR-61 membranes equilibrated with different concentrations of MgCl₂ solution, in addition to the S *1s* binding energy and corresponding Donnan potential values, are provided in Supplementary Figs. 12 and 13. The magnitude of the Donnan potential and the extent of co-ion exclusion depend strongly on counter-ion valence[17]. All other factors being equal, the Donnan potential is lower for membranes that have sorbed higher valence counter-ions, leading to reduced co-ion exclusion. On the other hand, a given Donnan potential will more strongly exclude co-ions of higher valence[17]. Remarkably, the experimental Donnan potential data is in good agreement with these heuristics from the historical literature. Consequently, the Donnan potential of CR-61 membranes equilibrated with MgCl₂ solutions is lower at each given external salt concentration compared to NaCl-equilibrated CR-61. Moreover, the concentration dependence of the Donnan potential is significantly weaker for MgCl₂-equilibrated CR-61 compared to NaCl-equilibrated CR-61. This behavior is also captured from the slopes of the linear fit of experimental data, where the slope of the NaCl curve is ~2 times that of MgCl₂, as predicted by the Donnan framework.

## Comparison between experimental data and theoretical model
To complement our APXPS data, we compared measured Donnan potential values to values predicted by using a thermodynamic model for ion partitioning in IEMs based on the Manning[27,28]/Donnan[15,16] theories. In many studies, including Donnan's original theory (i.e., the classic Donnan model), ion activity coefficients in the membrane and solution phases are often eliminated from analysis by assuming either ideal behavior (i.e., ion activity coefficients of unity) or equality of ion activity coefficients in the membrane and solution[15,16]. These assumptions are recognized as inconsistent with electrolyte and polyelectrolyte thermodynamic theories[29,30]. For this reason, we applied Manning's counter-ion condensation theory to predict ion activity coefficients in CR-61 membranes and calculated ion activity coefficients in the external salt solutions using the Pitzer model[31]. This approach (i.e., the Manning/Donnan model) uses no adjustable parameters, requiring only basic properties of the membrane (i.e., water

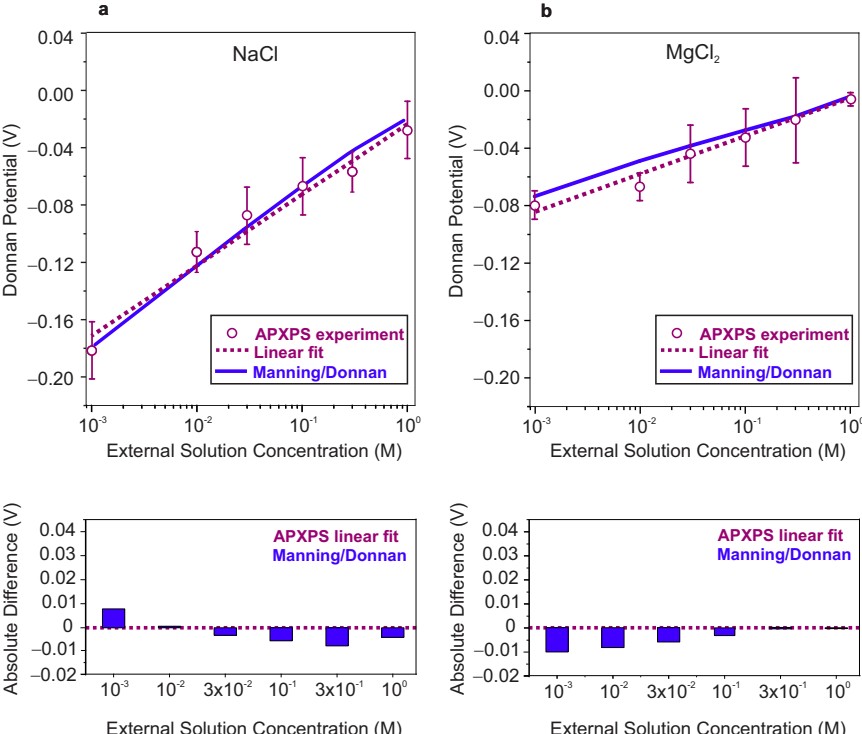

**Fig. 3 | Comparison between the experiment and theory.** Comparison of experimentally measured Donnan potential values obtained from binding energy shifts in membrane related core levels with the Manning/Donnan model predictions as a function of external **a** NaCl and **b** $MgCl_2$ solution concentration. Error bars represent experimental uncertainty and are the standard deviation of repeated measurements of a minimum of four different locations in each case. Dashed lines are the linear fits of ambient pressure X-ray photoelectron spectroscopy (APXPS) experimental data for external NaCl and $MgCl_2$ solutions with slopes of

$0.049 \pm 0.004$ and $0.026 \pm 0.002$, respectively. (Brief discussion and further analysis of experimental uncertainties and statistical significance are provided in Supplementary Note 7). Details on Donnan potential calculations using the Manning/Donnan models are given in Supplementary Note 8. The absolute difference between the linear fit of the experimental data and predictions from the analytical model is provided to better capture the comparison between theory and experiment for each electrolyte concentration.

and fixed charge content). Details of these analytical calculations are provided in Supplementary Note 7. The Manning/Donnan model was previously applied to CR-61 membranes, with good agreement observed between this model's predictions and experimental ion sorption data[19,26]. Although this approach has been the focus of continued studies in the last several years[32], these studies have provided only comparisons of theory and experiment in terms of ion sorption (i.e., macroscopic data for solution and membrane phase concentrations and activity coefficients). Figure 3a, b compare the experimental and predicted Donnan potential values as a function of external NaCl and $MgCl_2$ solution concentrations, respectively. Comparisons of experimental results with predictions using the classic Donnan model are also presented in Supplementary Note 9, with a detailed discussion. To evaluate the agreement between the experimental values and the theoretical model, we calculated the absolute difference between the linear fit of the experimental data and model predictions. Donnan potential values for NaCl equilibrated CR-61 predicted using the Manning/Donnan model (Supplementary Note 8) agree remarkably well with the APXPS experiments, over the entire concentration range probed, despite the lack of adjustable parameters in this model. This observation is consistent with previous studies of ion exchange polymers and polyelectrolyte solutions, which report low ion activity coefficients that increase slightly with increasing external salt concentration[26,29,30]. For $MgCl_2$ equilibrated CR-61 (Fig. 3b), Manning/Donnan predictions of the Donnan potential exhibit relatively larger deviations (although still within experimental uncertainty) from the experimental data at low external salt concentrations (0.001–0.03 M). Deviations from theoretical predictions may reflect a breakdown of one or more assumptions of the models. For instance, Manning's

counter-ion condensation theory considers only local electrostatic interactions and neglects those between different polymer chains or between distant fixed charges on the same chain. This limiting law derived by Manning[28] may not be valid at low external salt concentrations where electrostatic interactions between different chains may be significant, due to the very small amount of mobile salt sorbed in the polymer. At higher external salt concentrations, Donnan potential values predicted using the Manning/Donnan model are in better agreement with the experimental data for $MgCl_2$ equilibrated CR-61. These results suggest that the Donnan potential values measured using our methodology are consistent, at least within experimental uncertainty, with current thermodynamic models used in the membrane literature. In summary, these results provide direct evidence of the Donnan potential, validating the utility of this historical framework, and highlight the importance of measuring the Donnan potential to inform the development of more comprehensive thermodynamic models for charged polymers equilibrated with electrolyte solutions.

In conclusion, using tender-APXPS, we have directly measured the Donnan potential at the membrane/solution interface for CR-61 membranes equilibrated with NaCl and $MgCl_2$. The binding energy shifts in the core level peaks of the membrane as a function of the external salt concentration are directly related to the Donnan potential. This direct measurement was heretofore assumed to be impossible. We believe that this work represents a satisfactory proof-of-principle for a method that has general applicability and is not restricted to one specific class of IEMs or electrolyte solutions. Demonstration of more complex membrane–electrolyte systems, which could exhibit additional thermodynamic non-idealities or ion

specific effects, would be of great fundamental and practical interest. Our work in that direction is currently in progress. Our approach is particularly appealing because it provides a direct measurement of Donnan potential at the membrane/solution interface, unlike traditional transmembrane potential measurements that can provide only indirect information about the magnitude of Donnan potential for completely permselective membranes in the absence of diffusion potential[33,34]. Furthermore, we have compared our experimentally determined potentials with values predicted using well-established thermodynamic models, providing the first test of applicability of these models in estimating the Donnan potential. Future improvements in the experimental set-up will enhance spectral resolution and signal-to-noise ratio, which will improve our ability to discern the fine differences among various thermodynamic models. Our work along those lines is in progress. Finally, while existing thermodynamics models do not capture all molecular interactions and non-idealities, our approach provides a much-needed experimental method of gathering data to inform the development of new theoretical models to better describe ion behavior at the membrane/solution interface. The knowledge gained from this study has the potential to impact a wide range of scientific fields, including electrochemical conversion and storage of energy, water purification, and environmental and biological sciences.

## Methods

### Materials
Commercially available CR-61 styrenic cation exchange membranes were obtained from Suez in the form of $Na^+$ counter-ion. The chemical structure and other relevant membrane properties are given in Supplementary Table 1; the listed properties were used for theoretical calculations of the Donnan Potential. High-purity NaCl and $MgCl_2$ were obtained from Aldrich (99.9%) and used as received without further purification. MilliQ water (18.2 MOhm cm$^{-1}$) served as the solvent.

### Counter-ion exchange of CR-61
CR-61 cation exchange membranes were supplied in the $Na^+$ counter-ion form. Prior to APXPS measurements for $MgCl_2$ equilibrated samples, the membranes were converted to the $Mg^{2+}$ counter-ion forms through an ion exchange process. Water equilibrated membranes were soaked in a 0.1 M $MgCl_2$ solution for at least 24 h. Salt solutions were exchanged several times to ensure removal of all sodium counter-ions. The concentration of sodium was negligible after a few hours of soaking in the salt solution, indicating complete conversion to the divalent cation form, as seen in previous studies[19,35]. Following this ion exchange procedure, membranes were soaked in deionized water to desorb mobile salt. Deionized water was exchanged twice daily for 2 days, after which standard APXPS analysis of the prepared surfaces further confirmed that ion exchange to the $Mg^{2+}$ form was complete.

### Preparation of APXPS samples and "dip and pull" measurements
CR-61 ion exchange membranes were cut down to 1 cm×5 cm rectangles. Prior to measurements, the membranes were washed and equilibrated with MilliQ water. In order to fully remove impurities, water was periodically refreshed for at least a day. After that, CR-61 samples were placed in NaCl or $MgCl_2$ solutions (for experiments with $MgCl_2$ solutions, ion exchange was performed prior to this step) at desired salt concentrations, ranging from 0.001 to 1.0 M, for at least two days before APXPS measurements. Salt solutions were again periodically replaced to ensure equilibrium between the membrane and the solution of desired concentration. After equilibration with the solution, membranes were removed, and the surface of each sample was carefully wicked with KimWipes to ensure the removal of any solution not sorbed into the polymer matrix. Samples were then mounted on the manipulator of the instrument.

Before their introduction into the analysis chamber, fresh salt solutions were outgassed for at least an hour at low pressure (~7 Torr) in a dedicated preparation chamber. pH of salt solutions were measured to be circumneutral. Once the CR-61 sample and the outgassed salt solution were placed in the analysis chamber, the pressure was carefully lowered down to the water vapor pressure (~14–18 Torr) near room temperature. All the measurements were done in the absence of atmospheric $CO_2$ to eliminate the possibility of competitive ion interactions especially at lower solution concentrations. Before dipping samples into the solutions, we estimated water content inside the solution-equilibrated membranes in terms of the number of sorbed water molecules per sulfonate group. This estimate was calculated by obtaining the ratio of O $1s$ peak areas of LPW to sulfonate components, multiplied by a stoichiometric factor of 3, to be representative of the number of functional groups[36]. The "dip and pull"[21–24] procedure was carried out to obtain a stable electrolyte layer ~17–21 nm thick on the CR-61 membrane. During this procedure, the membrane was first fully immersed into the liquid and kept there at least an hour to ensure equilibrium between the two phases. The sample was then slowly extracted from the liquid by raising the sample manipulator. In this way, a thin layer of aqueous electrolyte film was formed on the membrane surface. The layer was then positioned at the intersection of the X-ray beam and the focal point of the hemispherical electron analyzer. By changing the height of the measurement spot, a suitable liquid thickness for the experiment was found. Then, the distance between the nozzle and sample was optimized to allow measurements of the solid/liquid interface. Fast O $1s$ core level spectra were collected at the beginning and end of each acquisition cycle to assess water loss due to evaporation. Measurements with unstable water content were disregarded. To estimate experimental uncertainties, statistical analyses were performed on repeated measurements of a minimum of five different spots at each concentration of solution. These measurement spots were chosen by pulling the membrane up at least 1 mm (i.e., larger than the X-ray spot size) and were verified, by visual inspection of the areal ratios of O1s core levels, to have similar liquid layer thicknesses at the same concentration. If the LPW was lower (i.e., drier membrane), the dip and pull process was repeated to obtain complete wetting. Since the membrane sample length is known and positions are recorded, we ensured to always select a fresh analysis location and never measured the same physical location of the membrane twice. The liquid film thickness depends on a number of factors including: (i) the hydrophilicity of the membrane surface, (ii) the height of the measurement spot above the free surface of the bulk liquid, and (iii) solution concentration.

### Tender APXPS data collection and data treatment
All XPS measurements were conducted in the tender APXPS Beamline (B.L. 9.3.1) at Lawrence Berkeley National Laboratory's Advanced Light Source (ALS). This instrument is equipped with tender X-Rays having energy range between 2.0 and 6.0 keV[21]. All core level spectra reported here were taken using 4.0 keV photon energy in normal emission and 100 eV analyzer (R4000 HiPP-2, Scienta) pass energy. With 4.0 keV photon energy, the inelastic mean free path (IMFP; $\lambda$) of the O $1s$ core level corresponds to 9.5 nm, which makes the XPS probing depth ~28.5 nm (from $3\lambda$). During APXPS measurements, pressure in the analysis chamber ranged from 14 to 18 Torr, while the analyzer was kept under high vacuum conditions (~4 × $10^{-7}$ Torr). The calibration of the BE scale was performed using the bulk LPW photoelectron peak from the external electrolyte solution. (Details of the energy calibration process are given in Supplementary Note 4) All the fits reported in this work were carried out using Casa XPS software and a symmetrical Gaussian/Lorentzian product function after Shirley background subtraction. Because the membrane was chosen specifically to avoid beam-induced damage on the polymer surfaces, no chemistry change was observed throughout the experiment. In addition, ion-exchange

membranes are highly ion conductive under hydrated conditions. Thus, we observed no charging throughout the APXPS measurements in this study. To estimate the experimental uncertainties, we performed statistical analyses of repeated measurements of 4–5 different positions at each concentration. Error bars were determined from the standard deviation. To check experimental reproducibility, we repeated the measurements on CR-61 membranes equilibrated with the lowest and highest concentrations of salt solution on different days.

### Donnan potential calculations

Theoretically determined Donnan potential values are derived from the condition of thermodynamic equilibrium by treating the IEM matrix as a continuous phase with homogeneously distributed fixed charges[16]. At equilibrium, the electrochemical potential of the ions in the membrane ($\tilde{\mu}_i^m$) and the solution ($\tilde{\mu}_i^s$) phases are assumed equal, as follows:

$$\tilde{\mu}_i^m = \tilde{\mu}_i^s = \mu_i^{m0} + RT \ln a_i^m + F z_i \Psi^m = \mu_i^{s0} + RT \ln a_i^s + F z_i \Psi^s \qquad (1)$$

where $F$ is the Faraday constant, $z_i$ is the valence of the ion, $a_i$ is the activity of the ion ($a_i = \gamma_i C_i$), and superscripts m and s stand for the membrane and electrolyte solution, respectively. The Donnan potential ($\Psi_D$) is expressed as the potential difference between the membrane $\Psi^m$ and the solution $\Psi^s$ potentials and can be related to the activity ratios of either the co-ions or counter-ions:

$$\Psi_D = \Psi^m - \Psi^s = \frac{RT}{z_i F} \ln \frac{\gamma_i^s C_i^s}{\gamma_i^m C_i^m} \qquad (2)$$

where $T$ is the absolute temperature, $R$ is the gas constant, $\gamma_i^m$ and $\gamma_i^s$ are the ion activity coefficients, and $C_i^m$ and $C_i^s$ are the concentrations of counter-ions or co-ions in the membrane and solution phase, respectively. The dependence of the Donnan potential on pressure has not been directly included in this expression (i.e., the hydrostatic pressures in the membrane and solution phases are assumed to be equal), given that previous studies show that this simplification has a negligible effect on modeling ion equilibria in IEMs[17,19,37]. Ion concentrations in the solution were determined using previously reported experimental ion sorption data. In the classic Donnan model, ion activity coefficients in the membrane and solution are eliminated by assuming equality of ion activity coefficients in each phase[16,17]. The Manning/Donnan model, on the other hand, uses Manning's counter-ion condensation theory to predict ion activity coefficients in CR-61 membranes, while ion activity coefficients in the external salt solutions were calculated using the Pitzer model[31]. Details of these calculations are provided in Supplementary Note 8.

### Numerical simulation of potential drop at membrane/solution interface

To simulate the electrical potential distribution across a charged membrane in equilibrium with an electrolyte solution, we used the model previously presented by Ohshima et al.[38], which assumes that the electrical potential in the solution region and membrane region satisfies the Poisson–Boltzmann equation. Numerical solutions of equations showing potential distribution as a function of position in planar coordinates were completed in Python. The information needed for numerically simulating the potential distribution includes temperature ($T$), charge ($z$), relative solution ($\epsilon_r$) and the membrane ($\epsilon_r'$) permittivity, counter-ion concentrations inside the membrane ($C_g^m$), and the external salt solution concentration ($C_g^s$). Details of the simulation are given in Supplementary Note 2.

### Simulation of XPS core level spectra from calculated potential drop

All numerical simulations of the XPS core level spectra were finished in Python. Three pieces of information are needed for numerically simulating the spectra: binding energy (BE), which determines the center of the individual peak; Gaussian broadening, which is obtained from the experimental full width at half maximum (FWHM); and peak area, which is determined by the number density of each element and integrating over the exponential escape probability of the photoelectron intensity (Beer-Lambert Law)[39,40].

By assuming that the 1:1 shift in the binding energy location following the electrical potential profile (see Fig. 1c) is a function of membrane distance from the interface[4] and the exponentially decaying intensity $e^{-(z\lambda^{-1})}$ (where $z$ is the probing depth of the membrane with respect to the interface and $\lambda$ is the inelastic-mean-free-path of the photoelectron), we can sum up the different spectral contributions with a fine grid of 0.1 Å to obtain S $1s$ and O $1s$ spectra directly comparable with our experiments (Supplemetary Note 3).

## Data availability

The binding energy and area of processed data are provided in the Supplementary Information file. The remaining experimental data related to this work, including the raw and processed spectra, are available upon request from the corresponding author. Source data are provided with this paper.

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

## Acknowledgements

This work was supported as part of the Center for Materials for Water and Energy Systems (M-WET), an Energy Frontier Research Center funded by the U.S. Department of Energy, Office of Science, Basic Energy Sciences under Award #DE-SC0019272. This research used resources of the Advanced Light Source, which is a DOE Office of Science User Facility, under contract no. DE-AC02-05CH11231. J.Q., Y.W., and E.J.C. were partially supported by an Early Career Award in the Condensed Phase and Interfacial Molecular Science Program, in the Chemical Sciences Geosciences and Biosciences Division of the Office of Basic Energy Sciences of the U.S. Department of Energy, under Contract No. DE-AC02-05CH11231.

## Author contributions

The manuscript was written with contributions from all authors. P.A.G. planned and executed the experiments, evaluated the data and drafted the manuscript. E.J.C. supervised the study, planned the experiments and drafted the manuscript. R.S. executed the sorption experiments. P.A.G., R.S., J.Q., and Y.W. performed the theoretical calculations. L.K. and B.F. gave collaborative support, engaged in discussions, and actively participated in drafting the manuscript. All authors have given approval to the final version of the manuscript.

## Competing interests

The authors declare no competing interests.
