## [Peer Review File · Nature Communications]

The Donnan Potential RevealedREVIEWER COMMENTS

Reviewer #1 (Remarks to the Author):

This manuscript describes tender XPS measurements aimed at determining Donnan potentials. These sophisticated experiments are carefully thought out and provide a direct method for determining Donnan potentials, with some theoretical corrections applied at low salt concentrations. Prior methods could only assess differences in Donnan potentials, and even then with some caveats. The manuscript is well-written and should be clear and of interest to those in the field of ion transport through biological and separation membranes. It is nice to see a direct measurement of the Donnan potential.

The main challenge in the technique is that because the differences in binding energies are small, the error bars in determinations of Donnan potential are typically $\pm 20\text{-}30\%$ or even more. The uncertainties in these measurements are often larger than deviations from theoretical calculations. Thus, I am not convinced that this technique is appropriate for testing theories, especially small effects. Some of the authors already demonstrated the improved modelling of ion-exchange membranes with a Manning/Donnan model, so that is not new. The manuscript overstates differences between calculated and measured potentials. I recommend publication with major revisions because the direct measurement of Donnan potentials is certainly satisfying. However, the authors should reconsider their data and whether discrepancies between theory and experiment are statistically significant.

1. In contrast to the tone of the manuscript, my thought after seeing the data was that the simple Donnan model fits these data surprisingly well. It essentially predicts the Donnan potential to within the error bars of the XPS data, particularly for MgCl_2 . On page 12, the manuscript states, "For MgCl_2 equilibrated CR-61 (Fig. 3b), both the classic Donnan and Manning/Donnan predictions of the Donnan potential exhibit relatively large deviations from the experimental data at low external salt concentrations (0.001-0.03 M)." The deviations are smaller than the typical error bars in the measurements.

2. I am surprised the simple Donnan model fits the MgCl_2 data so well (essentially within experimental error). I would have thought that strong electrostatic interactions between Mg^{2+} and negative fixed charge would have caused more deviations. Can the authors explain this good fit in one or two sentences?

3. A transmembrane potential measurement using different solutions on the two sides of the membrane should provide a difference in Donnan potentials and indicate how Donnan potentials vary with concentration. Such measurements have challenges when diffusion potentials inside the membrane become significant, and one needs to account for changes in reference-electrode potentials when varying solution compositions. Nevertheless, with inexpensive instrumentation, transmembrane potentials provide data related to those mentioned here. A few sentences contrasting the techniques are needed.

4. Some of the text in Figure 1 is too small to read at normal magnifications.

5. Are there limits on how low of a salt concentration one can use? Is low solution conductivity a problem, and could it affect the determined Donnan potential? If a low salt concentration is not a problem, why not go to lower concentrations to get a much larger change in binding energy? Is the membrane grounded via the solution? I cannot tell if this is the case in Figure 1 and would appreciate some comment on whether this is important.

6. In Figure 3b, why are the error bars so much smaller for the first and last points? Is this simply a random event? Perhaps there are better statistical methods to treat the uncertainties using pooled values.

7. On page 17, the authors write "The dependence of Donnan potential on pressure has not been considered, given previous studies showing that it has little, if any, effect on the ion sorption." I would appreciate a reference or two to substantiate this statement.

8. The authors note that the probing depth is ~28.5 nm for the O 1s level. Does the layer of water on the surface lead to strong attenuation of the S 1s signal, and is this an issue in the measurements? I would appreciate a brief comment on this in the supplementary information.
9. Is there a reason that the study did not examine the XPS spectra of the counterions in the membrane to give further verification of the Donnan potential? A brief comment in the supplementary information would suffice.
10. The manuscript discusses double-layer calculations to correct the O 1s line from liquid-phase water. In looking at Figure 1a, it appears that the O 1s spectra are independent of the ionic strength. Does this suggest that the double-layer correction is superfluous, or are these spectra already corrected? The supplementary information discusses this, but it is not clear if Figure 1 is already corrected.
11. In supplementary Figure 5, the binding energies on the axes in Figure 5b and 5c are not the same. Additionally, to the eye it appears there is no change in the different spectra in Figure 5b, especially because there is only one line in the figure.
12. In supplementary Table 3, how are the authors defining the double-layer thicknesses calculated using the Poisson-Boltzmann equation?

Reviewer #2 (Remarks to the Author):

Understanding the membrane/liquid interface has important implications in biological membranes non-biological membrane technologies. This paper used synchrotron-based ambient pressure X-ray photoelectron spectroscopy with 4 keV X-rays in combination with the dip-pull method to examine O 1s and S 1s binding energy shifts of the ion exchange membrane CR-61 as a function of bulk salt solution concentration. The shifts were then examined against model contributions from the electric double layer (EDL) and Donnan potential at the membrane/liquid interface. This manuscript is very well written and easy (even enjoyable) to follow. The magnitude of the change in measured slopes as presented in Figure 3 appear to nicely correspond to the change in valence in Equation S8, with finer details being assessed beyond the classical Donnan model.

While these results are interesting and significant, measuring binding energy shifts at solid/solution interfaces is not very transformative; especially for this beamline with a proof-of-concept paper in 2015 (ref 12). Moreover, this paper falls short in experimental novelty and depth of analysis compared to the 2016 Favaro et al. Nature Communications paper from this group (ref 13).

The following are comments/questions that should be addressed. Experimentally, the results being presented are too simplistic, assessing only O 1s and S 1s positions in this manuscript. From a practical perspective, it would be surprising if O 1s and S 1s spectra were the only two spectra collected for each concentration. That would be an unproductive use of valuable beamtime. This is a carbon-based material. The assessment of the carbon spectra would monitor potential changes in chemistry and the potential influences of adventitious carbon. At the very least, the authors should present and discuss corresponding C 1s data if it was collected. If it wasn't, why not? The normality (eq/L) of Na⁺ and Mg²⁺ is the same as the sulfonate group in the membrane, and should therefore be measurable. Were Na⁺ and Mg²⁺ measured? If not, why not? Measuring the counter ion would have been a very interesting complement to the S 1s data, especially if different behavior was observed. Also, representative XPS survey data are needed to show that no other chemistry

is present in the system. The suggested homogeneously distributed 20 nm film of salt solution across the polymer surface (Fig 2b,c) is not very convincing. XPS is not structural, and an attenuation calculation is not proof of this. How does the 20 nm compare to the polymer surface roughness? Were the five difference spots examined via APXPS different both horizontally and vertically? It should be made clear how each particular spot was chosen, their relative distance from each other and why. For example, was a particular spot chosen because certain peak areas of GPW/LPW/membrane are seen in the O 1s? Were there any spots nearby that did not have significant LPW? How the vertical z-alignment was chosen should be explained as well, which determines the gas phase signal. The data in Figure 3 should also be presented using the raw data (not the averages at each concentration) and the slope +/- standard error given for both NaCl and MgCl₂ plots. From this analysis, are the slopes statistically different from each other? I would suggest "2 standard errors" as being significant. Is the MgCl₂ slope statistically different from zero? The determination of the overall (very small) peak shifts from XPS spectral fits was done without providing detailed fitting procedures. This is a bit concerning. The specific methods used to arrive at the determined peak areas, FWHM and positions need to be given, including any constraints used for all fits. The raw numbers also should be tabulated and made available in a table in the SI. Additional fits to the data for all concentrations should also be presented in the SI for visual inspection. This manuscript focuses just on peak shifts, but there should also be peak area analyses. If the LPW film is indeed homogeneous, the ratio of LPW O 1s area to membrane S 1s area should be consistent across all spots. Present this in a graph in the SI for all concentrations with some discussion. Also, the membrane O 1s area to membrane S 1s should be consistent. Present this as well. The shifting (or lack thereof) has not been discussed for O 1s GPW BE and membrane BE. The membrane O 1s BE should be consistent with the S 1s. Is it? It would be particularly interesting to see if the GPW shifts (i.e., EDL of solution/gas interface) with salt concentration, although if no significant trend is seen this would not be surprising for the non-polarizable chloride salts. Such a simple analysis of gas phase signal is not beyond the scope of this investigation, and can be presented in the SI and mentioned in a sentence or two in the main manuscript. Moreover, presenting these other shifts (O 1s GPW & membrane) will present an independent noise level of BE changes as a function of salt concentration. Finally, the SI needs to be reordered such that it is written in the order presented in the main text.

Reviewer #3 (Remarks to the Author):

The manuscript describes an experimental technique, tender-APXPS, to quantitatively understand the Donnan potential at the membrane/electrolyte interface. The binding energy shifts in the core level was correlated to the electric potential drop across the membrane/electrolyte interface, and two types of salts, NaCl and MgCl₂ were used to evaluate the Donnan potential drop as well the correlation to the theoretical models. The reviewer recommends the publication of this manuscript with some minor modification. It would be helpful to expand the discussion of the dip and pull method use in this study and its implication and application to understand the real system where the electrolyte layer thickness is much larger; can author also discuss the case where HCl or H₂SO₄ are used as the electrolyte at different concentrations and their corresponding Donnan potential as well as any measurement complications? The core energy level shifts with regard to the Donnan potential should have a linear relation with a slope of 1, which seems to be the case based on Figure 2c, and it would be helpful to indicate that.

The Donnan Potential Revealed

Pinar Aydogan Gokturk¹, Rahul Sujana², Jin Qian^{1,3}, Ye Wang^{1,3}, Lynn Katz⁴, Benny D. Freeman² and Ethan J. Crumlin^{1,3*}

¹ Advanced Light Source, Lawrence Berkeley National Laboratory, Berkeley, CA 94720, United States

² McKetta Department of Chemical Engineering, The University of Texas at Austin, Austin, Texas 78712, United States

³ Chemical Sciences Division, Lawrence Berkeley National Laboratory, Berkeley, California 94720, United States

⁴ Department of Civil, Architectural, and Environmental Engineering, The University of Texas at Austin, Austin, Texas 78712, United States

Dear Editors and Reviewers,

Thank you for the careful evaluations of our manuscript and providing the detailed comments, questions and suggestions. All the comments are very valuable and have been helpful for improving our manuscript.

We have addressed each comment very carefully, and have revised our manuscript accordingly. All changes in the manuscript are **highlighted in yellow**. The original comments, questions and queries, and our responses are in *italics and in blue color* can be found below

Reviewer(s)' Comments to Author:

Reviewer: 1

Comments:

This manuscript describes tender XPS measurements aimed at determining Donnan potentials. These sophisticated experiments are carefully thought out and provide a direct method for determining Donnan potentials, with some theoretical corrections applied at low salt concentrations. Prior methods could only assess differences in Donnan potentials, and even then with some caveats. The manuscript is well-written and should be clear and of interest to those in the field of ion transport through biological and separation membranes. It is nice to see a direct measurement of the Donnan potential.

The main challenge in the technique is that because the differences in binding energies are small, the error bars in determinations of Donnan potential are typically $\pm 20-30\%$ or even more. The uncertainties in these measurements are often larger than deviations from theoretical calculations. Thus, I am not convinced that this technique is appropriate for testing theories,

especially small effects. Some of the authors already demonstrated the improved modelling of ion-exchange membranes with a Manning/Donnan model, so that is not new. The manuscript overstates differences between calculated and measured potentials. I recommend publication with major revisions because the direct measurement of Donnan potentials is certainly satisfying. However, the authors should reconsider their data and whether discrepancies between theory and experiment are statistically significant.

Author reply: We are grateful that the reviewer has realized the value of our work and has also raised very probing questions. We agree that the primary aim of this manuscript was to show the first direct measurement of Donnan Potential rather than testing thermodynamic theories. We have significantly revised the manuscript to emphasize this point and believe this revision, based on the reviewer's comments, has significantly improved our manuscript. Specific detailed changes are provided below.

1. In contrast to the tone of the manuscript, my thought after seeing the data was that the simple Donnan model fits these data surprisingly well. It essentially predicts the Donnan potential to within the error bars of the XPS data, particularly for MgCl₂. On page 12, the manuscript states, "For MgCl₂ equilibrated CR-61(Fig. 3b), both the classic Donnan and Manning/Donnan predictions of the Donnan potential exhibit relatively large deviations from the experimental data at low external salt concentrations (0.001-0.03 M)." The deviations are smaller than the typical error bars in the measurements.

Author reply: see below, where we answer the reviewer's questions 1 & 2 together.

2. I am surprised the simple Donnan model fits the MgCl₂ data so well (essentially within experimental error). I would have thought that strong electrostatic interactions between Mg²⁺ and negative fixed charge would have caused more deviations. Can the authors explain this good fit in one or two sentences?

Author reply: We thank the reviewer for their careful consideration of our work. Many studies in the membrane community have focused on comparing experimental ion sorption data (i.e., ion concentrations in IEMs) with theoretical predictions using either the classic Donnan or the Manning/Donnan model. This motivated our comparison of Donnan potential measurements with these models. However, when applying these models to describe the Donnan potential, differences between models can become mathematically smaller because the calculations are proportional to the natural logarithm of the counter-ion concentration ratio between each phase. This is especially evident for MgCl₂ equilibrated CR-61, where the Donnan potential values are relatively small due to fewer counter-ions being present in the membrane relative to the NaCl case. Moreover, as the external salt concentration increases and the counter-ion concentrations in each phase become similar, the electrostatic interactions between the fixed ions and counter-ions are screened, and the thermodynamic environment of the membrane becomes similar to that of the solution. For these reasons, membrane phase activity coefficients of divalent counter-ions become quite similar to their values in the solution phase at high external salt concentrations, as reported by Galizia et al., and the classic Donnan assumption becomes more reasonable. Thus for MgCl₂, the predictions of both models can be considered reasonable within the current error bars of our measurement.

In addition, as the reviewer pointed out, our motivation was not to test the accuracy of different thermodynamic models, but to show the first direct measurement of the Donnan potential using APXPS. Thus, we have revised our manuscript to compare our experimental results only with the Manning/Donnan model, because it has already been demonstrated to be a more accurate thermodynamic model for ion-exchange membranes, and to highlight our focus on measuring the Donnan potential. The discussion about the classic Donnan model and comparisons have been moved to the Supplementary File, with the additional points in reference to MgCl_2 presented there as well.

Fig. 3, Page 11-12 and Supplementary Note 5 have been revised as follows (changes made are highlighted in yellow)

In the Manuscript:

Fig. 3 Comparison between the experimental and predicted Donnan potential trends, values, and the effect of counter-ion valence on the Donnan potential. Comparison of experimentally measured Donnan potential values obtained from binding energy shifts in membrane related core levels with the Manning/Donnan model predictions as a function of external a, NaCl and b, MgCl₂ solution concentration. Error bars represent experimental uncertainty and are the standard deviation of repeated measurements of a minimum of four different positions in each case. Dashed lines are the linear fits of APXPS experimental data for external NaCl and MgCl₂ solutions with slopes of 0.049 ± 0.004 and 0.026 ± 0.002 , respectively.

(Brief discussion and further analysis of experimental uncertainties and statistical significance are provided in Supplementary Note 6.) Details on Donnan potential calculations using the Manning/Donnan models are given in Supplementary Note 7. The absolute difference between the linear fit of the experimental data and predictions from the analytical model is provided to better capture the comparison between theory and experiment for each electrolyte concentration.

Comparison between experimental data and theoretical model

To complement our APXPS data, we compared measured Donnan potential values to values predicted by using a thermodynamic model for ion partitioning in IEMs based on the Manning^{33,34}/Donnan^{23,24} theories. In many studies, including Donnan's original theory (i.e., the classic Donnan model), ion activity coefficients in the membrane and solution phases are often eliminated from analysis by assuming either ideal behavior (i.e., ion activity coefficients of unity) or equality of ion activity coefficients in the membrane and solution^{23,24}. These assumptions are recognized as inconsistent with electrolyte and polyelectrolyte thermodynamic theories^{35,36}. For this reason, we applied Manning's counter-ion condensation theory to predict ion activity coefficients in CR-61 membranes and calculated ion activity coefficients in the external salt solutions using the Pitzer model³⁷. This approach (i.e., the Manning/Donnan model) uses no adjustable parameters, requiring only basic properties of the membrane (i.e., water and fixed charge content). Details of these analytical calculations are provided in Supplementary Note 7. The Manning/Donnan model was previously applied to CR-61 membranes, with good agreement observed between this model's predictions and experimental ion sorption data^{28,33}. Although this approach has been the focus of continued studies in the last several years³⁸, these studies have provided only comparisons of theory and experiment in terms of ion sorption (i.e., macroscopic data for solution and membrane phase concentrations and activity coefficients). Fig. 3a and 3b compare the experimental and predicted Donnan potential values as a function of external NaCl and MgCl₂ solution concentrations, respectively. Comparisons of experimental results with predictions using the classic Donnan model are also presented in Supplementary Note 8, with a detailed discussion. To evaluate the agreement between the experimental values and the theoretical model, we calculated the absolute difference between the linear fit of the experimental data and model predictions. Donnan potential values for NaCl equilibrated CR-61 predicted using the Manning/Donnan model (Supplementary Note 7) agree remarkably well with the APXPS experiments, over the entire concentration range probed, despite the absence of any adjustable parameters in this model. This observation is consistent with previous studies of ion exchange polymers and polyelectrolyte solutions, which report low ion activity coefficients that increase slightly with increasing external salt concentration^{32,35,36}. For MgCl₂ equilibrated CR-61 (Fig. 3b), Manning/Donnan predictions of the Donnan potential exhibit relatively larger deviations (although still within our experimental uncertainty) from the experimental data at low external salt concentrations (0.001-0.03 M). Deviations from theoretical predictions may reflect a breakdown of one or more assumptions of the models: for instance, Manning's counter-ion condensation theory considers only local electrostatic interactions and neglects those between different polymer chains or between distant fixed charges on the same chain. This limiting law derived by Manning³⁴ may not be valid at low external salt concentrations where electrostatic interactions between different chains may be significant, due to the very small amount of mobile salt sorbed in the polymer. At higher external salt concentrations, Donnan potential values predicted using the Manning/Donnan model are in better agreement with the experimental data for MgCl₂ equilibrated CR-61. These results suggest that the Donnan potential values measured using our methodology are consistent, at least within experimental uncertainty, with current thermodynamic models used in the membrane literature. In summary, these results provide direct evidence of the Donnan potential, validating the utility of this historical framework, and

highlight the importance of measuring the Donnan potential to inform the development of more comprehensive thermodynamic models for charged polymers equilibrated with electrolyte solutions.

In the Supporting Information:

Supplementary Note 8.

Comparison between experimental data and classical Donnan model

Supplementary Fig. 12a and 12b compare the experimental and classical Donnan model predicted Donnan potential values as a function of external NaCl and MgCl₂ solution concentrations, respectively. To assess the agreement between the experimental values and the theoretical model, we calculated the absolute difference from the linear fit of the experimental data. For NaCl equilibrated CR-61 (**Supplementary Fig. 12a**), agreement between the classic Donnan model and experimental data is rather poor at low external solution concentrations, but as the external salt concentration increases, agreement between the theoretical model and experimental measurements improve. The inability of the classic Donnan model to accurately predict the Donnan potential arises, at least in part, from strong electrostatic interactions between fixed charge groups and counter-ions, leading to the condensation of counter-ions on the polymer backbone. This results in highly non-ideal thermodynamic behavior, especially at low electrolyte concentrations. In contrast, when an IEM is equilibrated with a highly concentrated solution, the fixed charge groups can be electrostatically screened by sorbed salt, and the thermodynamic environments of the solution and membrane phases become more similar. In this way, the ion activity coefficients in the membrane and in the external electrolyte solution also become closer to one another, such that the classic Donnan model can provide a reasonable prediction of the Donnan potential in CR-61 equilibrated with 1 M NaCl.

For MgCl₂ equilibrated CR-61 (**Supplementary Fig. 12b**), predictions using the classic Donnan model correlate reasonably well with the experimental measurements. We note that for MgCl₂ equilibrated CR-61, fewer counter-ions are present in the membrane at any given external salt concentration, leading to relatively low Donnan potential values. Moreover, previous studies of CR-61 have shown that activity coefficients in the membrane are more similar to activity coefficients in the external solution for CaCl₂ equilibrated CR-61 than for NaCl equilibrated CR-61, particularly at higher external salt concentrations.²³ We believe that the ability of the classic Donnan model to reasonably predict the Donnan potential, within the error of our measurements, arises from the similar behavior of MgCl₂ equilibrated to previously reported CaCl₂ equilibrated CR-61.

Supplementary Fig. 12. Comparison between the experimental and predicted Donnan potential trends, values, and the effect of counter-ion valence on Donnan potential. Comparison of experimentally measured Donnan potential values obtained from binding energy shifts in membrane related core levels with the classic Donnan model predictions as a function of external **a**, NaCl and **b**, MgCl₂ solution concentration. Error bars represent experimental uncertainty and are the standard deviation of repeated measurements of a minimum of five different positions in each case. Dashed lines are the linear fits of APXPS experimental data for external NaCl and MgCl₂ solutions with slopes of 0.049 and 0.026, respectively. (Details on Donnan potential calculations using the classic Donnan models appear in **Supplementary Note 7**). The absolute difference between the linear fit of the experimental data and predictions from the analytical model is provided to better capture the comparison between theoretical model and experimental result for each electrolyte concentration.

3. A transmembrane potential measurement using different solutions on the two sides of the membrane should provide a difference in Donnan potentials and indicate how Donnan potentials vary with concentration. Such measurements have challenges when diffusion potentials inside the membrane become significant, and one needs to account for changes in reference-electrode potentials when varying solution compositions. Nevertheless, with inexpensive instrumentation, transmembrane potentials provide data related to those mentioned here. A few sentences contrasting the techniques are needed.

Author reply: We thank the reviewer for the suggestion. The potential difference between two solutions with different concentrations that are separated by a permselective membrane is referred to as the membrane potential. Contrary to the Donnan potential, the membrane potential can be measured directly in an electrolytic cell using reversible electrodes. The membrane potential is equal to the two different Donnan potentials established at each interface plus the diffusion potential caused by differences in the transport number of different ions in the membrane. Such transmembrane potential measurements for a completely permselective membrane, where the diffusion potential is zero, could provide information about the Donnan Potential. However, this is an indirect method and membranes are not perfectly permselective, so a diffusion potential is commonly exhibited inside the membrane. In addition, as emphasized by the reviewer, these measurements are challenging because one needs to account for changes in reference-electrode potentials when varying solution compositions. On the other hand, the method we presented can easily be expanded to more complex systems and used to understand the diffusion potentials from the broadening of XPS peak shape. Future work along these lines is in progress.

Page 13 has been revised to include relevant discussion (changes made are highlighted in yellow) as follows:

“In conclusion, using tender-APXPS, we have directly measured the Donnan potential at the membrane/solution interface for CR-61 membranes equilibrated with NaCl and MgCl₂. We have shown that the binding energy shifts in the core level peaks of the membrane as a function of the external salt concentration are directly related to the Donnan potential. This direct measurement was heretofore assumed to be impossible. We believe that this work represents a satisfactory proof-of-principle for a method that has general applicability and is not restricted to one specific class of IEMs or electrolyte solutions. Demonstration of more complex membrane-electrolyte systems, which could exhibit additional thermodynamic non-idealities or ion specific effects, would be of great fundamental and practical interest. Our work in that direction is currently in progress. This approach is particularly appealing because it provides a direct measurement of Donnan potential at the membrane/solution interface, unlike ordinary transmembrane potential measurements that can provide only indirect information about the magnitude of Donnan potential for completely permselective membranes in the absence of diffusion potential.^{9,39} Furthermore, we have compared our experimentally determined potentials with values predicted using well-established thermodynamic models, providing the first test of applicability of these models in estimating the Donnan potential. Future improvements in the experimental set-up will enhance spectral resolution and signal-to-noise ratio, which will improve our ability to discern the fine differences among various thermodynamic models. Our work along those lines is in progress. Finally, it is well known that existing thermodynamics models do not capture all molecular interactions and non-idealities, but our approach provides a much-needed experimental method of gathering data to inform the development of new theoretical models that better describe the behavior of ions at the

membrane/solution interface. The knowledge gained from this study has the potential to impact a wide range of scientific fields, including electrochemical conversion and storage of energy, water purification, and environmental and biological sciences.”

4. Some of the text in Figure 1 is too small to read at normal magnifications.

Author reply: We thank our reviewer sharing this detailed comment. We modified the corresponding texts in Figure 1 as suggested.

5. Are there limits on how low of a salt concentration one can use? Is low solution conductivity a problem, and could it affect the determined Donnan potential? If a low salt concentration is not a problem, why not go to lower concentrations to get a much larger change in binding energy? Is the membrane grounded via the solution? I cannot tell if this is the case in Figure 1 and would appreciate some comment on whether this is important.

Author reply: We would like to thank the reviewer for raising these questions. We selected our concentration range considering the salinity scale in various water sources (e.g. fresh water, brackish water, saline water and brine) that are relevant to purification applications. In theory, there are no lower limits on the electrolyte concentration, however using concentrations lower than 1mM introduces practical difficulties in maintaining the desired concentration. Contamination of the solution by low levels of ions on the equipment can become an issue. Although it is not apply for our experimental set-up, some prior works show that competitive ion sorption by other ions in solution (e.g., from CO₂ speciation) also becomes important at low concentrations. It introduces error unless one accurately account for the impact on ionic strength or the impact associated with competitive adsorption. Preparing the solutions and making the measurements in a CO₂-free environment become a necessary precaution. With sufficient care and effort to mitigate these challenges, measurements at lower salt concentrations can be achieved.

Although the ion-exchange membranes are highly ion conductive in hydrated conditions, most of them, including CR-61, are electrical insulators. Thus, none of the system components were grounded. However, it should be emphasized that no charging was observed throughout the

APXPS measurements in this study. Further details can be found in Supplementary Fig. 2; the energy level diagram of the membrane/solution/vapor system is included below in the original form for the reviewer's convenience.

Supplementary Fig. 2. Energy level diagram of a membrane/solution/vapor system during the *operando* tender-APXPS measurements **a**, when there is no concentration difference between counter-ion species in the solution and the membrane phase (i.e. $\Psi_{Donnan} = 0$ V) and **b**, when there is a concentration gradient between the solution and membrane phase (i.e., concentration of counter-ions in solution is lower than the number of fixed ions in the membrane). **c**, Corresponding binding energy shifts of photoelectron spectra originating from each components of the system. Solid line represents peaks under the potential gradient and dashed-lines indicate peaks when there is no concentration difference between counter-ion species in the solution and the membrane phase (electrical potential is equal to 0 V). The electrical potential developed on the membrane effectively shifts all energy levels up (with negative potential) relative to the grounded analyzer. The measured binding energy of membrane related core level photoelectrons show a shift that is related to the Donnan potential (Ψ_{Donnan}) on the surface by $\Delta BE = \Psi_{Donnan}$ eV ($h\nu$: incoming X-ray energy, E_F : Fermi level, E_V : vacuum level, CL : core level, $\Phi_{spectrometer}$: spectrometer work function, KE : photoelectron kinetic energy, BE : binding energy).

6. In Figure 3b, why are the error bars so much smaller for the first and last points? Is this simply a random event? Perhaps there are better statistical methods to treat the uncertainties using pooled values.

Author reply: We thank the reviewer for raising this question. The smaller error bars for the lowest and highest concentrations are random. Note that the experiment for each concentration was performed on separate days, and synchrotron beam stability during the experiment can vary, perhaps giving rise to these differences. However, we avoided using pooled values because the wetting of the membrane, the thickness, and the stability of the liquid layer depend on the concentration of the electrolyte solution. Unstable or thinner liquid layer formation introduces further experimental challenges and increases the uncertainties of our measurements.

7. On page 17, the authors write “The dependence of Donnan potential on pressure has not been considered, given previous studies showing that it has little, if any, effect on the ion sorption.” I would appreciate a reference or two to substantiate this statement.

Author reply: We thank the reviewer for this suggestion; we have included relevant references (ref 26, 28 and 42) in the revised manuscript. It is common to treat the hydrostatic pressure of the membrane and the external solution as being equal (cf. Kamcev et al and Helfferich). Doing so simplifies mathematical derivations considerably and has been shown to negligibly impact thermodynamic calculations of ion partitioning in ion exchange resins (cf. Mackie and Meares).

8. The authors note that the probing depth is ~28.5 nm for the O 1s level. Does the layer of water on the surface lead to strong attenuation of the S 1s signal, and is this an issue in the measurements? I would appreciate a brief comment on this in the supplementary information.

Author reply: Supplementary Note 1 has been revised to include details about the attenuation of the photoelectron signal as follows.

In the Supplementary Information:

“The main challenge of the membrane/liquid study with APXPS is related to the short inelastic mean free paths (IMFPs) of photoelectrons, the very same characteristic that makes the technique surface sensitive. In our experimental setup, membrane related core level photoelectrons (i.e., S 1s, C 1s and O 1s-SO₃) are attenuated through both the liquid and the gas phase. Creating a liquid film that is sufficiently stable and robust to be representative of a realistic membrane/solution interface at equilibrium, but thin enough to allow signal detection from the interfacial region, is very important. Considering these factors, a liquid layer of appropriate thickness was formed on the membrane.

With an estimated liquid layer ~17-21 nm thickness, the intensity of the O 1s peak for the SO₃⁻ component decreases by ~88% of its initial (dry) value due to attenuation through the liquid layer, which requires more collection time to get sufficient signal to noise of the spectra.”

9. Is there a reason that the study did not examine the XPS spectra of the counterions in the membrane to give further verification of the Donnan potential? A brief comment in the supplementary information would suffice.

Author reply:

We thank our reviewer for asking this question. We did collect the counter-ion specific core levels during our experiments. Surprisingly, they showed different behavior than that observed in S1s core levels. Our later experiments showed that this behavior is specific to the type of counter-ion. We believe that extending this work to other counter-ions, to show counter-ion specific behaviors, will be an interesting and valuable contribution to the field. We look forward to sharing more about this phenomenon in a future manuscript that is currently being prepared and are currently devoting significant time and effort in this direction.

10. The manuscript discusses double-layer calculations to correct the O 1s line from liquid-phase water. In looking at Figure 1a, it appears that the O 1s spectra are independent of the ionic strength. Does this suggest that the double-layer correction is superfluous, or are these spectra already corrected? The supplementary information discusses this, but it is not clear if Figure 1 is already corrected.

Author reply: We would like to thank the reviewer for this comment. We revised the Figure 2 caption to include the information about double layer corrections as follows (changes made are highlighted in yellow):

“Fig. 2 Experimental measurements of the Donnan potential from spectral binding energy shifts of sulfonate group with respect to solution-related core level peaks. Representative **a**, O 1s and **b**, S 1s core level spectra collected from CR-61 membranes equilibrated with different concentrations of NaCl solutions. The binding energy is calibrated using bulk liquid phase water (LPW). The effects of EDL on the binding energy position of the LPW peak are considered, and corrections are made during the calibration process. (Details of the energy calibration process are given in Supplementary Note 4.) Circles represent raw experimental data, and lines indicate the sum of fits. Representative spectra not provided here for three other concentrations of NaCl solutions are presented in Supplementary Fig. 6 for visual inspection. **c**, Averaged S 1s binding energy and the corresponding Donnan potential values as a function of the external solution concentration. Note that the S1s binding energy shift versus Donnan potential shows a linear dependence of ~ 1 eV/V. Error bars that represent the experimental uncertainty were determined from the standard deviation of repeated measurements of a minimum of five different positions in each case. The binding energy value of fitted core level components for each individual analysis position are provided in Supplementary Table 6. The dashed line represents the best fit to a linear dependence. The fitted line was extrapolated to the equivalent concentration of counter-ions (3.2 M), where the Donnan potential approaches 0 V²⁵.”

11. In supplementary Figure 5, the binding energies on the axes in Figure 5b and 5c are not the same. Additionally, to the eye it appears there is no change in the different spectra in Figure 5b, especially because there is only one line in the figure.

Author reply: We updated Supplementary Fig. 8 (former Fig. 5) as follow:

Supplementary Fig. 9 Experimental measurements of the Donnan potential from spectral binding energy shifts of sulfonate groups with respect to solution related core level peaks. Representative **a**, O 1s and **b**, S 1s core level spectra collected from CR-61 membranes equilibrated with different concentrations of MgCl₂. Circles represent raw experimental data and lines indicate the sum of fits. Representative O 1s and S 1s core level spectra collected from CR-61 membranes equilibrated with three other concentrations of MgCl₂ solutions are presented in **Supplementary Fig 10**. **c**, Averaged S 1s binding energy and corresponding Donnan potential values as a function of the external solution concentration. Error bars represent experimental uncertainty, determined from the standard deviation of repeated measurements of four or more different positions in each case. The binding energy value of fitted core level components for each individual analysis position are provided in **Supplementary Table 3**. The dashed line represents a linear best fit. The fitted line was extrapolated to the equivalent concentration of counter-ions (1.5 M for MgCl₂) where the Donnan potential approaches to 0 V.

12. In supplementary Table 3, how are the authors defining the double-layer thicknesses calculated using the Poisson-Boltzmann equation?

Author reply: Electrical double layer thicknesses are estimated from the numerical solutions of the Poisson-Boltzmann equation showing potential distribution as a function of position. It can be defined as the overall length of a Gouy-Chapman type diffuse layer.

Supplementary Note 4 has been revised as follows:

“The EDL thicknesses for various external concentrations, estimated from the commonly used Debye-Huckel Theory¹² (linearized Poisson-Boltzmann equation) and the Poisson-Boltzmann simulations (**Supplementary Note 2**), are given in **Supplementary Table 7**. **Electrical double layer thickness from the numerical solutions of Poisson-Boltzmann equation are obtained from the overall length of Gouy-Chapman type diffuse layer in **Supplementary Fig. 5**.** As expected, the Debye length underestimates the EDL width. Corresponding BE shifts in LPW were estimated by simulating XP spectra from the potential distributions within the EDL region.”

Reviewer: 2

Comments:

Understanding the membrane/liquid interface has important implications in biological membranes non-biological membrane technologies. This paper used synchrotron-based ambient pressure X-ray photoelectron spectroscopy with 4 keV X-rays in combination with the dip-pull method to examine O 1s and S 1s binding energy shifts of the ion exchange membrane CR-61 as a function of bulk salt solution concentration. The shifts were then examined against model contributions from the electric double layer (EDL) and Donnan potential at the membrane/liquid interface. This manuscript is very well written and easy (even enjoyable) to follow. The magnitude of the change in measured slopes as presented in Figure 3 appear to nicely correspond to the change in valence in Equation S8, with finer details being assessed beyond the classical Donnan model.

While these results are interesting and significant, measuring binding energy shifts at solid/solution interfaces is not very transformative; especially for this beamline with a proof-of-concept paper in 2015 (ref 12). Moreover, this paper falls short in experimental novelty and depth of analysis compared to the 2016 Favaro et al. Nature Communications paper from this group (ref 13).

Author reply: We thank the reviewer for finding our work interesting and significant. We believe this revision, based on the reviewer's comments, has improved our paper significantly.

Experimentally, the results being presented are too simplistic, assessing only O 1s and S 1s positions in this manuscript. From a practical perspective, it would be surprising if O 1s and S 1s spectra were the only two spectra collected for each concentration. That would be an unproductive use of valuable beamtime. This is a carbon-based material. The assessment of the carbon spectra would monitor potential changes in chemistry and the potential influences of adventitious carbon. At the very least, the authors should present and discuss corresponding C 1s data if it was collected. If it wasn't, why not? The normality (eq/L) of Na⁺ and Mg²⁺ is the same as the sulfonate group in the membrane, and should therefore be measurable. Were Na⁺ and Mg²⁺ measured? If not, why not? Measuring the counter ion would have been a very interesting complement to the S 1s data, especially if different behavior was observed. Also, representative XPS survey data are needed to show that no other chemistry is present in the system.

Author reply: We thank the reviewer for the careful consideration of our manuscript. We wanted this work to attract researchers from various fields, especially from the membrane science community. Our aim was not simply to measure binding energy shifts at solid/solution interfaces but to show the first direct measurement of the Donnan potential of membranes, which was previously considered to be impossible. While APXPS techniques have been presented in prior publications, they have yet to be leveraged in the membrane science community or to investigate the Donnan potential in any community. For this reason, the manuscript was written to be easy to understand and follow, especially for those who are not familiar with the technique. However, we understand our reviewer's concerns about being simplistic and have revised the manuscript to address this concern. We consider this a success, as we were able to accomplish a complex feat previously deemed impossible and communicate it in a clear and approachable manner. As the body of work provided in this manuscript demonstrates, this is far from simple and requires great care and diligence to accomplish.

All core levels were collected during the experiments, including the C1s and counter-ion specific ones. In addition, our reviewer is right in that the Donnan potential of the CR-61 membrane equilibrated with an aqueous salt solution can also be assessed from the binding energy shifts of other membrane related core levels. The problem with the C1s region is that it consists of at least 5 different chemistries, including aromatic, aliphatic, and C-SO₃ associating from the membrane. In addition to the spectral features coming from the membrane itself, more C 1s chemistries became visible (in the form of adventitious carbon and other carbonaceous species i.e., C-O) upon exposure to water. Deconvolution of this peak is challenging since the many individual contributions forming the C 1s peak are not well resolved. We also could not constrain the area of any membrane components during the fitting procedure, due to the unknown percentage of cross linker in the membrane assembly. This made the peak-fitting process challenging and following such small binding energy differences in the region unreliable. For the counter-ions, we certainly agree with our reviewer that measurements on the counter-ion are an interesting complement to the S 1s data, especially because they show different behavior. Our later experiments with different counter-ions (not shown here) demonstrated that this behavior is counter-ion specific. We believe that extending this work to other counter-ions to show counter-ion specific behaviors will be an interesting and valuable contribution to the field. We are currently devoting time and effort to better understanding this phenomenon and hope to publish our results.

According to the suggestion of the reviewer, we revised our Supplementary Information to present the B.E. positions of all core levels, including other membrane related core levels, and included a discussion about why we particularly chose to follow the changes in the S 1s core level in the first place. Supplementary Table 6, presented below, is for the **reviewers only**. **The version of this table given in Supplementary file does not include the binding energies of counter-ions.**

In addition, all the raw data will be made available upon request and uploaded for reviewers who are interested.

In the Main text (Page 9):

“We note that the Donnan potential of the CR-61 membrane equilibrated with an aqueous salt solution can also be extracted from the binding energy shifts of other membrane related core levels (i.e., O 1s-membrane and C 1s). A discussion about other core levels is presented in Supplementary Note 5.”

In the Supplementary File:

Supplementary Table 6. Binding energy results of fitted core level peaks of membrane/solution system collected from four or five different positions for each electrolyte concentration

Name	P #1 B.E. (eV)	P #2 B.E. (eV)	P #3 B.E. (eV)	P #4 B.E. (eV)	P #5 B.E. (eV)
CR61- 1 mM NaCl					
O 1s-LPW	533.11	533.11	533.11	533.11	533.11
O 1s-GPW	535.28	535.30	535.25	535.28	535.33
O 1s- SO₃	531.59	531.62	531.60	531.58	531.60
S 1s-SO₃	2476.95	2476.96	2476.95	2476.92	2476.93
Na 1s	1071.62	1071.64	1071.63	1071.62	1071.64
CR61- 10 mM NaCl					
O 1s-LPW	533.11	533.11	533.11	533.11	533.11
O 1s-GPW	535.30	535.26	535.31	535.30	535.35
O 1s- SO₃	531.64	531.65	531.66	531.66	531.63
S 1s-SO₃	2477.00	2476.99	2476.98	2476.98	2476.97
Na 1s	1071.62	1071.62	1071.62	1071.66	1071.62
CR61- 30mM NaCl					
O 1s-LPW	533.11	533.11	533.11	533.11	533.11
O 1s-GPW	535.22	535.30	535.25	535.24	535.29

O 1s- SO₃	531.65	531.66	531.69	531.68	531.66
S 1s-SO₃	2476.98	2477.01	2477.01	2477.03	2477.00
Na 1s	1071.62	1071.64	1071.64	1071.64	1071.62
CR61- 0.1 M NaCl					
O 1s-LPW	533.11	533.11	533.11	533.11	533.11
O 1s-GPW	535.37	535.34	535.32	535.31	535.33
O 1s- SO₃	531.70	531.65	531.67	531.68	531.64
S 1s-SO₃	2477.00	2477.01	2477.00	2477.04	2477.03
Na 1s	1071.61	1071.60	1071.62	1071.63	1071.62
CR61- 0.3 M NaCl					
O 1s-LPW	533.11	533.11	533.11	533.11	533.11
O 1s-GPW	535.26	535.30	535.35	535.35	535.23
O 1s- SO₃	531.72	531.67	531.67	531.69	531.69
S 1s-SO₃	2477.04	2477.03	2477.02	2477.02	2477.04
Na 1s	1071.68	1071.68	1071.64	1071.62	1071.65
CR61- 1 M NaCl					
O 1s-LPW	533.11	533.11	533.11	533.11	533.11

O 1s-GPW	535.30	535.33	535.30	535.34	535.29
O 1s- SO₃	531.70	531.70	531.71	531.68	531.73
S 1s-SO₃	2477.07	2477.07	2477.05	2477.03	2477.06
Na 1s	1071.61	1071.68	1071.63	1071.63	1071.61
CR61- 1 mM MgCl₂					
O 1s-LPW	533.11	533.11	533.11	533.11	N/A
O 1s-GPW	535.26	535.15	535.21	535.24	N/A
O 1s- SO₃	531.60	531.60	531.63	531.62	N/A
S 1s-SO₃	2476.96	2476.96	2476.97	2476.97	N/A
Mg 1s	1304.53	1304.54	1304.54	1304.55	N/A
CR61- 10 mM MgCl₂					
O 1s-LPW	533.11	533.11	533.11	533.11	533.11
O 1s-GPW	535.23	535.19	535.20	535.24	535.18
O 1s- SO₃	531.64	531.65	531.63	531.61	531.63
S 1s-SO₃	2476.97	2476.99	2476.96	2476.99	2476.99
Mg 1s	1304.56	1304.52	1304.56	1304.53	1304.54
CR61- 30 mM MgCl₂					
O 1s-LPW	533.11	533.11	533.11	533.11	533.11

O 1s-GPW	535.25	535.28	535.19	535.22	535.23
O 1s- SO₃	531.67	531.67	531.64	531.67	531.63
S 1s-SO₃	2476.99	2477.00	2477.03	2476.98	2476.99
Mg 1s	1304.54	1304.53	1304.53	1304.57	1304.56
CR61- 0.1 M MgCl₂					
O 1s-LPW	533.11	533.11	533.11	533.11	533.11
O 1s-GPW	535.24	535.16	535.21	535.19	535.20
O 1s- SO₃	531.67	531.66	531.65	531.67	531.66
S 1s-SO₃	2477.01	2477.04	2477.02	2476.98	2477.01
Mg 1s	1304.56	1304.51	1304.50	1304.51	1304.54
CR61- 0.3 M MgCl₂					
O 1s-LPW	533.11	533.11	533.11	533.11	N/A
O 1s-GPW	535.14	535.11	535.16	535.17	N/A
O 1s- SO₃	531.65	531.67	531.65	531.66	N/A
S 1s-SO₃	2477.05	2477.03	2477.02	2476.99	N/A
Mg 1s	1304.52	1304.56	1304.59	1304.57	N/A
CR61- 1 M MgCl₂					

O 1s-LPW	533.11	533.11	533.11	533.11	533.11
O 1s-GPW	535.14	535.19	535.05	535.01	535.09
O 1s- SO₃	531.68	531.68	531.70	531.69	531.68
S 1s-SO₃	2477.04	2477.04	2477.04	2477.06	2477.04
Mg 1s	1304.50	1304.50	1304.48	1304.46	N/A

Supplementary Note 5.

Binding energy shifts of other core levels

In theory, the Donnan potential of CR-61 membranes equilibrated with aqueous salt solutions can also be assessed from the binding energy shifts of other membrane related core levels (i.e., O 1s-SO₃ and C 1s). **Supplementary Table 6** shows the binding energies of fitted XP spectra for O 1s-LPW, O 1s-GPW, O 1s-SO₃, S 1s-SO₃ and Na 1s. C 1s core level spectra are also collected during the experiments. However, the C 1s region consists of at least 5 different chemistries including aromatic, aliphatic, and C-SO₃ associating from the membrane. Additionally, the C 1s region is susceptible to further changes upon exposure to water (in the form of adventitious carbon and other carbonaceous species, i.e., C-O and C=O). Representative C 1s spectrum is provided in **Supplementary Fig. 7**. This makes the multi-peak fitting process challenging and following such small binding energy differences in the region unreliable. That is why we have not presented the binding energies of C 1s chemistries here.

As predicted, the binding energy shifts in membrane O 1s are consistent with those in S 1s (see **Supplementary Fig. 8 and Table 6**). However, the O 1s region requires multi-peak fitting and introduction of additional constraints on binding energy, which decreases accuracy of the measurement. In addition, due to the nature of this experimental technique, membrane related core level photoelectrons are attenuated through both the liquid and the gas phase. Low binding energy peaks shows relatively low photoelectron intensities due to their low cross sections. Considering these factors, in the main text, we chose a single component, high binding energy S 1s core level to measure the Donnan potential, leading to smaller error bars and more precise measurements.

On the other hand, no significant trend for binding energies of O 1s-GPW peak is observed with respect to the external solution concentration.

Supplementary Fig. 7 Representative C 1s core level spectra collected from CR-61 membranes equilibrated with 1M NaCl solution. The binding energy is calibrated using bulk liquid phase water (LPW). Circles represent the raw experimental data. Vertical dashed lines indicate possible individual contributions forming the C 1 s peak.

In addition, representative XPS survey spectra for CR61-Na and CR61-Mg are provided in Supplementary Fig. 1 to show that no other chemistry was present in the system.

In the Supplementary File:

Supplementary Fig. 1. Survey spectra of membrane/solution/vapor system during the *operando* tender-APXPS measurements for **a**, NaCl_(sol) equilibrated CR-61 and **b**, MgCl₂_(sol) equilibrated CR-61 membrane. No other chemistry was observed in the system.

The suggested homogeneously distributed 20 nm film of salt solution across the polymer surface (Fig 2b,c) is not very convincing. XPS is not structural, and an attenuation calculation is not proof of this. How does the 20 nm compare to the polymer surface roughness? Were the five difference spots examined via APXPS different both horizontally and vertically? It should be made clear how each particular spot was chosen, their relative distance from each other and why. For example, was a particular spot chosen because certain peak areas of GPW/LPW/membrane are seen in the O 1s? Were there any spots nearby that did not have significant LPW? How the vertical z-alignment was chosen should be explained as well, which determines the gas phase signal.

Author reply: We agree with the reviewer that XPS gives no structural information about the surface, and we didn't mean to suggest that we formed a homogeneous liquid layer on the polymer. Given the nature of the dip and pull method (forming a meniscus), it is not possible to

form this kind of layer with a specific thickness. Using the attenuation calculations, we were able only to estimate the thickness of the layer at the spot where data were collected. However, with careful consideration of all spots analyzed for various electrolyte concentration, we determined an approximate liquid layer thickness of ~17-21 nm (note that intensity ratios and equations needed for the relevant calculations have been provided in Supplementary Tables 3 and 4 and Note 1). Additionally, pages 16-17 have been revised as follows to include further experimental details requested by our reviewer (changes made are highlighted in yellow)

In the Main Text:

“The “dip and pull”^{11,12,29,30} procedure was carried out to obtain a stable electrolyte layer ~17-21 nm thick on the CR-61 membrane. During this procedure, the membrane is first fully immersed into the liquid and kept there at least an hour to ensure equilibrium between the two phases. The sample is then slowly extracted from the liquid by raising the sample manipulator. In this way, a thin layer of aqueous electrolyte film was formed on the membrane surface. The layer was then positioned at the intersection of the X-ray beam and the focal point of the hemispherical electron analyzer. By changing the height of the measurement spot suitable liquid thickness for the experiment was found. Then, the distance between the nozzle and sample was optimized to allow measurements of the solid/liquid interface. Fast O1s core level spectra were collected at the beginning and the end of each acquisition cycle to assess water loss due to evaporation. Measurements with unstable water content were disregarded. In order to estimate the experimental uncertainties, statistical analyses were carried out on repeated measurements of a minimum of five different spots at each concentration of solution. These measurement spots were chosen by pulling the membrane up at least 1 mm (larger than the X-Ray spot size) and were verified, by visual inspection of the areal ratios of O1s core levels, to have similar liquid layer thicknesses within the same concentration. If the LPW was lower (i.e., drier membrane), the dip and pull process was repeated to obtain complete wetting. Since the membrane sample length is known and positions are recorded we ensured to always select a fresh analysis location and never measured the same physical location of the membrane twice. Note that the liquid film thickness depends on a number of factors including (i) the hydrophilicity of the membrane surface, (ii) the height of the measurement spot above the free surface of the bulk liquid, and (iii) the concentration of solutions.”

The data in Figure 3 should also be presented using the raw data (not the averages at each concentration) and the slope +/- standard error given for both NaCl and MgCl₂ plots. From this analysis, are the slopes statistically different from each other? I would suggest “2 standard errors” as being significant. Is the MgCl₂ slope statistically different from zero?

Author reply: We thank our reviewer for paying attention to these details. We have presented the requested version of Figure 3 using raw data and included a discussion on statistical significance in the Supplementary File.

In the Supplementary Information:

Supplementary Note 6.

Experimental uncertainties and statistical significance

Since our approach to directly measuring the Donnan potential at the membrane/solution interface is based on following small binding energy shifts of membrane related core level XP spectra, we utilized a number of statistical methods to estimate the experimental uncertainties,

determining outlier data points and evaluating statistical significance of our data. As mentioned in the main text, the error bars used to represent experimental uncertainty are determined from the standard deviation of repeated measurements of 4-5 different positions in each case.

In addition, we employed the statistical “Q test” to determine if a single data value is an outlier in a distinct sample size. This test essentially calculates the ratio between the putative outlier’s distance from its nearest neighbor and the range of values. Upon comparing the calculated Q to the theoretical Q, one very large (2477.12 eV S1s binding energy for CR-61 equilibrated with 0.1M NaCl) and one very small (2476.93 eV S1s binding energy for CR-61 equilibrated with 0.1M NaCl) values were rejected from the data set by the Q-test with 95% confidence.

To demonstrate that the slopes of linear fit to our experimentally measured binding energies of S 1s peak are statistically different from zero and from each other for NaCl and MgCl₂ equilibrated CR-61 membranes, we ran a statistical t-test (inferential statistical test used to determine if there is a significant difference between the means of two groups) and an ANOVA (Analysis of Variance) analysis using Origin software. Further information regarding these statistical analyses can be found elsewhere¹³. The data in **Fig. 3** are presented using the raw data in **Supplementary Fig. 11**, and the related parameters from the statistical analysis are given in **Supplementary Tables 9** and **10** for NaCl and MgCl₂, respectively. Since the t and F values exceed the critical value of t¹³ and F¹³ at 95% confidence level for specific degrees of freedom (DF) and slopes are larger than 2 standard errors, we conclude that the slopes are statistically different from zero and from each other.

Supplementary Fig. 11. Comparison between the experimental and predicted Donnan potential trends, values, and the effect of counter-ion valence on Donnan potential. Comparison of experimentally measured Donnan potential values obtained from binding energy shifts in membrane related core levels with the Manning/Donnan model predictions as a function of external **a**, NaCl and **b**, MgCl₂ solution concentration. Dashed lines are the linear fits of APXPS experimental data averages for external NaCl and MgCl₂ solutions with slopes of 0.049 ± 0.004 and 0.026 ± 0.002 , respectively. The plots are the same as those in **Fig. 3**; we have included the raw data here as well to facilitate the discussion of statistical significance.

Supplementary Table 9 Apparent fit parameters and ANOVA results for the experimentally measured binding energy shifts in membrane related core levels of NaCl-equilibrated CR61

PARAMETERS	Value	Standard Error	t-Value	Prob> t	
Intercept	2477.06	0.007	331782.47	4.95151E-22	
Slope	0.049	0.004	11.28	3.51475E-4	
ANOVA	DF	Sum of Squares	Mean Square	F Value	Prob>F
Model	1	34.53004	34.53004	127.32879	3.51475E-4
Error	4	1.08475	0.27119		
Total	5	35.61479			

Supplementary Table 10 Apparent fit parameters and ANOVA results for the experimentally measured binding energy shifts in membrane related core levels of MgCl₂-equilibrated CR61

PARAMETERS	Value	Standard Error	t-Value	Prob> t	
Intercept	2477.04	0.004	641282.80	3.54775E-23	
Slope	0.0261	0.002	11.24	3.57243E-4	
ANOVA	DF	Sum of Squares	Mean Square	F Value	Prob>F
Model	1	0.00395	0.00395	126.26996	3.57242E-4
Error	4	1.25244E-4	3.1311E-5		
Total	5	0.00408			

The determination of the overall (very small) peak shifts from XPS spectral fits was done without providing detailed fitting procedures. This is a bit concerning. The specific methods used to arrive at the determined peak areas, FWHM and positions need to be given, including any constraints used for all fits. The raw numbers also should be tabulated and made available in a table in the SI. Additional fits to the data for all concentrations should also be presented in the SI for visual inspection.

Author reply: We thank our reviewer for this comment. We have revised the Supplementary Information to include a table (Supplementary Table 5) showing the peak fitting parameters used for this study. Line shapes and full-width-half-maxima (FWHM) constraints are provided for each component of related core levels. Binding energies of fitted spectra from each analysis spot are given in Supplementary Table 6. Representative fits to the data for other concentrations have also been presented in the SI for visual inspection.

In the Supplementary Information:

Supplementary Table 5. Peak fitting parameters and constrains used for this study. Peak assignments were made by assuming the minimum number of peaks necessary to fit spectral features. Binding energies (BE – maxima and minima reported) and full-width-half-maxima (FWHM) agree well with values reported in literature^{2,5,6}

	O1s-GPW	O1s-LPW	O1s- SO ₃	S1s-SO ₃
Line Shape	GL(50-70)	GL(30)	GL(30)	GL(30)
FWHM	0.7-0.9	1.6-1.9	1.2-1.4	1.4-1.7
Binding Energy ^a	535.20-535.40	533.11	531.55-531.75	-

^aBinding energy values of fitted core level peaks are provided in Supplementary Table 6

Supplementary Fig. 6. Experimental measurements of spectral binding energy shifts of sulfonate group with respect to solution-related core level peaks. Representative **a**, O 1s and **b**, S 1s core level spectra collected from CR-61 membranes equilibrated with the three other concentrations of NaCl solutions that are not presented in the main text. The binding energy is calibrated using bulk liquid phase water (LPW). The effect of the double layer on binding energy position of LPW peak is considered and corrections are made during the calibration process. (Details of the energy calibration process are given in **Supplementary Note 4**.) Circles represent the raw experimental data, and lines indicate the sum of fits. The binding energy values of fitted core level components for other individual analysis positions are provided in **Supplementary Table 6**.

Supplementary Fig. 10. Experimental measurements of spectral binding energy shifts of sulfonate group with respect to solution-related core levels. Representative **a**, O 1s and **b**, S 1s core level spectra collected from CR-61 membranes equilibrated with other three concentrations of MgCl₂ solutions (not presented in the main text). The binding energy is calibrated using bulk liquid phase water (LPW). Circles represent the raw experimental data, and lines indicate the sum of fits. The binding energy value of fitted core level components for each individual analysis position are provided in **Supplementary Table 6**.

This manuscript focuses just on peak shifts, but there should also be peak area analyses. If the LPW film is indeed homogeneous, the ratio of LPW O 1s area to membrane S 1s area should be consistent across all spots. Present this in a graph in the SI for all concentrations with some discussion.

Author reply: We thank our reviewer for raising this question. We have tabulated the ratio of LPW O 1s area to membrane S 1s area in the Supplementary Information. We used these ratios to estimate the approximate thickness of the liquid layer on the analysis spots and some discussion has been added to Supplementary Note 1,

In the Supplementary Information:

Supplementary Table 3. Relative intensities of the solution and membrane related O 1s (LPW and SO_3^- components) XPS peaks of NaCl equilibrated CR-61 system collected from five different positions for each electrolyte concentration.

	P #1	P #2	P #3	P #4	P #5
1 mM NaCl	8.2	7.4	7.6	9.3	9.5
10 mM NaCl	9.8	8.2	7.2	7.2	7.2
30 mM NaCl	7.5	7.9	7.1	7.6	7.0
0.1 M NaCl	8.4	7.3	8.9	7.9	7.6
0.3 M NaCl	7.0	9.6	7.2	7.3	7.5
1 M NaCl	7.4	7.3	8.2	9.3	8.5

Supplementary Table 4. Relative intensities of the solution and membrane related O 1s (LPW and SO_3^- components) XPS peaks of MgCl_2 equilibrated CR-61 system collected from five different positions for each electrolyte concentration.

	P #1	P #2	P #3	P #4	P #5
1 mM MgCl_2	8.5	7.8	8.3	7.8	N/A
10 mM MgCl_2	7.2	7.0	7.0	7.4	8.3
30 mM MgCl_2	9.8	11.6	10.1	9.5	8.4
0.1 M MgCl_2	9.7	9.0	8.6	7.5	8.5
0.3 M MgCl_2	11.6	10.8	11.0	12.4	N/A
1 M MgCl_2	8.2	10.8	12.3	10.9	11.8

Although the relative intensities of the solution and membrane related O 1s peaks show fluctuations from spot to spot, they are largely very similar in terms of the actual thickness of the film. For the lowest ratio obtained for NaCl equilibrated CR-61 system (i.e., relative intensity is equal to 7.0) the thickness of liquid layer is calculated as 17 nm, whereas for the largest ratio obtained for NaCl equilibrated CR-61 system (i.e., relative intensity = 9.8) the calculated thickness of the liquid layer is 20 nm. Similarly, for the lowest ratio obtained for MgCl_2 equilibrated CR-61 system, the calculated thickness of the liquid layer is 17 nm, vs. 21 nm for the largest ratio obtained for MgCl_2 equilibrated CR-61. Thus, although the relative intensities of the solution and membrane related O 1s (LPW and SO_3^- components) XPS peaks of the MgCl_2 equilibrated membrane are higher (see **Supplementary Table 3 and 4**), the thickness of the liquid layer was estimated to be similar to those equilibrated with NaCl.

The shifting (or lack thereof) has not been discussed for O 1s GPW BE and membrane BE. The membrane O 1s BE should be consistent with the S 1s. Is it? It would be particularly interesting to see if the GPW shifts (i.e., EDL of solution/gas interface) with salt concentration, although if no significant trend is seen this would not be surprising for the non-polarizable chloride salts. Such a simple analysis of gas phase signal is not beyond the scope of this investigation, and can be presented in the SI and mentioned in a sentence or two in the main manuscript. Moreover, presenting these other shifts (O 1s GPW & membrane) will present an independent noise level of BE changes as a function of salt concentration.

Author reply: We thank our reviewer for raising these questions. Supplementary Information has been revised to present the B.E. position of all core levels, including O1s-LPW and O1s-SO₃ (see Supplementary Table 6). From the binding energies, it has been shown the shifts in membrane O 1s BEs are consistent with those in S 1s (Supplementary Fig. 8). However, no significant trend is observed for the GPW shifts. The relevant discussion is provided in Supplementary Note 5.-Binding energy shifts of other core levels

In the Supplementary Information:

Supplementary Note 5.

Binding energy shifts of other core levels

In theory, the Donnan potential of CR-61 membranes equilibrated with aqueous salt solutions can also be assessed from the binding energy shifts of other membrane related core levels (i.e., O 1s-SO₃ and C 1s). **Supplementary Table 6** shows the binding energies of fitted XP spectra for O 1s-LPW, O 1s-GPW, O 1s-SO₃, S 1s-SO₃ and Na 1s. C1s core level spectra are also collected during the experiments. However, the C1s region consists of at least 5 different chemistries including aromatic, aliphatic, and C-SO₃ associating from the membrane. Additionally, the C 1s region is susceptible to further changes upon exposure to water (in the form of adventitious carbon and other carbonaceous species, i.e., C-O and C=O). Representative C1s spectrum is provided in **Supplementary Fig. 7**. This makes the multi-peak fitting process challenging and following such small binding energy differences in the region unreliable. That is why we have not presented the binding energies of C1s chemistries here.

As predicted, the binding energy shifts in membrane O 1s are consistent with those in S 1s (see **Supplementary Fig. 8 and Table 6**). However, the O 1s region requires multi-peak fitting and introduction of additional constraints on binding energy, which decreases accuracy of the measurement. In addition, due to the nature of this experimental technique, membrane related core level photoelectrons are attenuated through both the liquid and the gas phase. Low binding energy peaks shows relatively low photoelectron intensities due to their low cross sections. Considering these factors, in the main text, we chose a single component, high binding energy S 1s core level to measure the Donnan potential, leading to smaller error bars and more precise measurements.

On the other hand, no significant trend for binding energies of O 1s-GPW peak is observed with respect to the external solution concentration.

Supplementary Fig. 8 Comparison of experimentally measured binding energy shifts in membrane related core levels as a function of external **a**, NaCl and **b**, MgCl₂ solution concentration.

Finally, the SI needs to be reordered such that it is written in the order presented in the main text

Author reply: We thank our reviewer for paying attention to the details. We have reordered the SI as much as possible without disrupting its overall flow

Reviewer: 3

Comments:

The manuscript describes an experimental technique, tender-APXPS, to quantitatively understand the Donnan potential at the membrane/electrolyte interface. The binding energy shifts in the core level was correlated to the electric potential drop across the membrane/electrolyte interface, and two types of salts, NaCl and MgCl₂ were used to evaluate the Donnan potential drop as well the correlation to the theoretical models. The reviewer recommends the publication of this manuscript with some minor modification.

Author reply: We thank our reviewer for evaluating our manuscript and finding it publishable in Nature Comm. after minor modifications. We have addressed each comment very carefully and have revised our manuscript accordingly.

It would be helpful to expand the discussion of the dip and pull method use in this study and its implication and application to understand the real system where the electrolyte layer thickness is much larger; can author also discuss the case where HCl or H₂SO₄ are used as the electrolyte at different concentrations and their corresponding Donnan potential as well as any measurement complications?

Author reply: We thank our reviewer for raising these questions. We have included proper references for the reader who wants further information about the advantages and limitations of the dip and pull method (refs 12, 13, 30 and 31). Our approach to measuring Donnan potential has general applicability and is not restricted to one specific class of membranes or electrolytes. It can be expanded to systems where acidic (i.e. HCl or H₂SO₄) or basic solutions (i.e., NaOH) are used as the electrolyte at different concentrations. However, we note that the degree of ionization of the functional groups of ion exchange membranes can depend on the environmental pH. While this is of limited concern for membranes with strongly acidic groups (e.g., sulfonated polymers) like those considered in this study, it would be an important consideration for polymers containing weakly acidic or basic groups (e.g., carboxylates). In addition, when the membranes are in acid-form, ionization of functional groups is affected by the amount of water sorbed. Thus, the dip and pull process should be done carefully to ensure complete hydration of the membrane.

Pages 12-13 have been revised as follows to include further details on the applicability of the technique, as requested by our reviewer (all changes made are highlighted in yellow):

“In conclusion, using tender-APXPS, we have directly measured the Donnan potential at the membrane/solution interface for CR-61 membranes equilibrated with NaCl and MgCl₂. We have shown that the binding energy shifts in the core level peaks of the membrane as a function of the external salt concentration are directly related to the Donnan potential. This direct measurement was heretofore assumed to be impossible. We believe that this work represents a satisfactory proof-of-principle for a method that has general applicability and is not restricted to one specific class of IEMs or electrolyte solutions. Demonstration of more complex membrane-electrolyte systems, which could exhibit additional thermodynamic non-idealities or ion specific effects, would be of great fundamental and practical interest. Our work in that direction is currently in progress. This approach is particularly appealing because it provides a direct measurement of Donnan potential at the membrane/solution interface, unlike ordinary

transmembrane potential measurements that can provide only indirect information about the magnitude of Donnan potential for completely permselective membranes in the absence of diffusion potential.^{9,39} Furthermore, we have compared our experimentally determined potentials with values predicted using well-established thermodynamic models, providing the first test of applicability of these models in estimating the Donnan potential. Future improvements in the experimental set-up will enhance spectral resolution and signal-to-noise ratio, which will improve our ability to discern the fine differences among various thermodynamic models. Our work along those lines is in progress. Finally, it is well known that existing thermodynamics models do not capture all molecular interactions and non-idealities, but our approach provides a much-needed experimental method of gathering data to inform the development of new theoretical models that better describe the behavior of ions at the membrane/solution interface. The knowledge gained from this study has the potential to impact a wide range of scientific fields, including electrochemical conversion and storage of energy, water purification, and environmental and biological sciences.”

The core energy level shifts with regard to the Donnan potential should have a linear relation with a slope of 1, which seems to be the case based on Figure 2c, and it would be helpful to indicate that.

Author reply: We thank our reviewer for paying attention to the details. Figure 2 caption in Page 8 has been revised to indicate this detail as follows (related changes made are highlighted in yellow):

“Fig. 2 Experimental measurements of the Donnan potential from spectral binding energy shifts of sulfonate group with respect to solution-related core level peaks. Representative **a**, O 1s and **b**, S 1s core level spectra collected from CR-61 membranes equilibrated with different concentrations of NaCl solutions. The binding energy is calibrated using bulk liquid phase water (LPW). The effects of EDL on the binding energy position of the LPW peak are considered, and corrections are made during the calibration process. (Details of the energy calibration process are given in Supplementary Note 4.) Circles represent raw experimental data, and lines indicate the sum of fits. Representative spectra not provided here for three other concentrations of NaCl solutions are presented in Supplementary Fig. 6 for visual inspection. **c**, Averaged S 1s binding energy and the corresponding Donnan potential values as a function of the external solution concentration. **Note that the S1s binding energy shift versus Donnan potential shows a linear dependence of ~ 1 eV/V.** Error bars that represent the experimental uncertainty were determined from the standard deviation of repeated measurements of a minimum of five different positions in each case. The binding energy value of fitted core level components for each individual analysis position are provided in Supplementary Table 6. The dashed line represents the best fit to a linear dependence. The fitted line was extrapolated to the equivalent concentration of counter-ions (3.2 M), where the Donnan potential approaches 0 V²⁵.”

REVIEWER COMMENTS

Reviewer #1 (Remarks to the Author):

The authors have given a detailed and appropriate response to my comments so I recommend publication.

Reviewer #2 (Remarks to the Author):

See pdf, which includes figures.

Reviewer #3 (Remarks to the Author):

The authors have adequately addressed my concerns.

Second review for Nature Communications (Nature) manuscript # NCOMMS-21-40448A.

Comments to authors.

Author reply:

We thank the reviewer for finding our work interesting and significant. We believe this revision, based on the reviewer's comments, has improved our paper significantly.

Reviewer reply:

Below the reviewer responds to some of the author replies with additional concerns.

Author reply:

All core levels were collected during the experiments, including the Cls and counter-ion specific ones. In addition, our reviewer is right in that the Donnan potential of the CR-61 membrane equilibrated with an aqueous salt solution can also be assessed from the binding energy shifts of other membrane related core levels.

Reviewer reply:

Based on the authors response, during experiments they collected S 1s, O 1s, C 1s, Na 1s, Mg 1s and Survey spectra during their beamtime. What about the chloride co-ion? Was this measured in the ~2800 eV BE region for Cl 1s? The survey provided stops at ~2500 eV. Were survey taken out further to capture Cl 1s as well? If data are available, they should be presented and discussed along with counter ions, as discussed by reviewer below.

Author reply:

The problem with the Cls region is that it consists of at least 5 different chemistries, including aromatic, aliphatic, and C-SO₃ associating from the membrane. In addition to the spectral features coming from the membrane itself, more C 1s chemistries became visible (in the form of adventitious carbon and other carbonaceous species i.e., C-O) upon exposure to water. Deconvolution of this peak is challenging since the many individual contributions forming the C 1s peak are not well resolved. We also could not constrain the area of any membrane components during the fitting procedure, due to the unknown percentage of cross linker in the membrane assembly. This made the peak-fitting process challenging and following such small binding energy differences in the region unreliable.

Reviewer reply:

The relative BE's of carbon moieties are well characterized in the XPS literature, as evidenced from the added Figure S7. The membrane examined is a carbon-based material, and the C 1s data collected should be presented and discussed beyond a "representative" spectrum. Specifically, the reviewer is wondering if there are also carbonate species present, which is often the case for APXPS work under wet conditions. Charge corrected C 1s spectra should be made available in the SI for the reader by creating simple stacked C 1s plots (similar to Figure S6, but for all conditions where C 1s was captured), one for NaCl and one for MgCl₂ solutions. And for these two plots, use vertical dashed lines indicating moiety specific BE's (similar to Figure S7) and provide references for the BE's used.

Author reply:

For the counter-ions, we certainly agree with our reviewer that measurements on the counter-ion are an interesting complement to the S 1s data, especially because they show different behavior. Our later experiments with different counter-ions (not shown here) demonstrated that this behavior is counter-ion specific. We believe that extending this work to other counter-ions to show counter-ion specific behaviors will be an interesting and valuable contribution to the field. We are currently devoting time and effort to better understanding this phenomenon and hope to publish our results.

Author reply:

The reviewer is concerned that the authors do not present or discuss counter-ion (or co-ion, if available) results in the manuscript. Using the data provided (for reviewers only), the reviewer created plots for Na 1s and Mg 1s BE shifts versus bulk concentration below. No shift is observed for Na 1s. The same is true for Mg 1s up through 100 mM, with an interesting drop in BE at 1000 mM. This should be presented and discussed in light of the clear shifts observed for immobile membrane peaks. The spectra should also be available for the reader, creating simple stacked plots for all concentrations, one for Na 1s, one for Mg 1s, and one for Cl 1s (if available). A question that arises for the reviewer is, did the authors observe more than one peak for counter-ions or co-ion? For example, are there bulk and membrane specific peaks observed above a certain salt concentration? If the answer is yes, then what BE is being reported in plots below? If the answer is no, then suggestions as to why would be interesting. Moreover, the reviewer would like to see a quantitative assessment of Na 1s/S 1s area ratio vs bulk concentration, Mg 1s/S 1s area ratio vs bulk concentration, and Cl 1s/S 1s (if available) vs. bulk concentration in order to assess potentially interesting adsorption properties. Only counter-ion BE's were provided to reviewers, not areas. This simple analysis makes Figure 1b more than just a cartoon. XPS can actually measure quantitatively the species present, and will further validate the Donnan model being put forth. If the membrane peaks can be separated from bulk peaks, these peak area ratios of membrane species vs. bulk solution concentration should be Langmuirian in nature.

Author reply:

In the Main text (Page 9):

“We note that the Donnan potential of the CR-61 membrane equilibrated with an aqueous salt solution can also be extracted from the binding energy shifts of other membrane related core levels (i.e., O 1s-membrane and C 1s). A discussion about other core levels is presented in Supplementary Note 5.”

Reviewer reply:

The revised discussion in quotes above makes no mention of the counter ions. The authors do not give a valid justification for this, and should include results and discussion of counter ion data as discussed by the reviewer above. Under revised Supplementary Note 5, Na 1s is mentioned at the end of the following sentence: “Supplementary Table 6 shows the binding energies of fitted XP spectra for O 1s-LPW, O 1s-GPW, O 1s-SO₃, S 1s-SO₃ and Na 1s.” The reviewer assumes this is a typo given the authors did not provide Na 1s data in Supplementary Table 6.

Author reply

In addition, representative XPS survey spectra for CR61-Na and CR61-Mg are provided in Supplementary Fig. 1 to show that no other chemistry was present in the system.

Reviewer reply:

These results are nice, showing no additional chemistry beyond what is expected. Are these results the same for all concentrations and experimental conditions? If so, an explicit statement should be made in the caption of Fig. S1 that says “No other chemistry was observed in the system *for all experiments*”. Should the S 1s label near ~230 eV be S 2s? Finally, if survey were collected out to Cl 1s (~2800 eV), this should also be shown.

Additional Reviewer comments:

- It is unclear why Figure S6 is repeated in Figure S10.

The Donnan Potential Revealed

Pinar Aydogan Gokturk¹, Rahul Sujamani², Jin Qian^{1,3}, Ye Wang^{1,3}, Lynn Katz⁴, Benny D. Freeman² and Ethan J. Crumlin^{1,3*}

¹ Advanced Light Source, Lawrence Berkeley National Laboratory, Berkeley, CA 94720, United States

² McKetta Department of Chemical Engineering, The University of Texas at Austin, Austin, Texas 78712, United States

³ Chemical Sciences Division, Lawrence Berkeley National Laboratory, Berkeley, California 94720, United States

⁴ Department of Civil, Architectural, and Environmental Engineering, The University of Texas at Austin, Austin, Texas 78712, United States

Dear Editor and Reviewers,

Thank you for the careful evaluations of our manuscript and providing the detailed comments, questions and suggestions. All the comments are very valuable and have been helpful for improving our manuscript.

We have addressed each comment very carefully, and have revised our manuscript accordingly. All changes in the manuscript are **highlighted in yellow**. The original comments, questions and queries, and our responses are in *italics and in blue color* can be found below

Reviewer(s)' Comments to Author:

Reviewer: 1

Comments:

The authors have given a detailed and appropriate response to my comments so I recommend publication.

Author reply: We thank our reviewer for evaluating our manuscript and finding it publishable in Nature Communications.

Reviewer: 2

Comments:

Based on the author's response, during experiments they collected S 1s, O 1s, C 1s, Na 1s, Mg 1s and Survey spectra during their beamtime. What about the chloride co-ion? Was this measured in the ~2800 eV BE region for Cl 1s? The survey provided stops at ~2500 eV. Were

survey taken out further to capture Cl 1s as well? If data are available, they should be presented and discussed along with counter ions, as discussed by reviewer below.

Author reply: We thank the reviewer for raising this important question. Unfortunately, survey spectra weren't taken out to capture the Cl 1s peak. However, during our beamtime, we collected high resolution spectra of Cl 1s core level from at least one location to see if it is detectable. No Cl 1s peak was observed for electrolyte concentrations lower than 1M. Considering the nature of ion exchange membranes, where co-ion concentration inside the membrane is negligible while counter-ion concentration is dictated by the number of fixed charges, we believe Cl 1s peaks were coming mainly from co-ions in the bulk solution which was below the detection limit of our instrument up to a certain salt concentration (e.g. 1 M)

The relative BE's of carbon moieties are well characterized in the XPS literature, as evidenced from the added Figure S7. The membrane examined is a carbon-based material, and the C 1s data collected should be presented and discussed beyond a "representative" spectrum. Specifically, the reviewer is wondering if there are also carbonate species present, which is often the case for APXPS work under wet conditions. Charge corrected C 1s spectra should be made available in the SI for the reader by creating simple stacked C 1s plots (similar to Figure S6, but for all conditions where C 1s was captured), one for NaCl and one for MgCl₂ solutions. And for these two plots, use vertical dashed lines indicating moiety specific BE's (similar to Figure S7) and provide references for the BE's used.

Author reply: We thank the reviewer for raising this question. We agree with the reviewer that presence of carbonate species is important especially for this type of experiment as competitive ion sorption by other ions in solution (e.g., carbonates from CO₂ speciation) could introduce error in the Donnan Potential measurement. No obvious carbonate species were detected in the C 1s spectra. As suggested by reviewer, Supplementary Fig.7 has been revised to include charge corrected C 1s spectra for all NaCl and MgCl₂ solution concentration, using vertical dashed lines indicating moiety specific BE's with the relevant references.

In the Supplementary Information:

Representative C1s spectra for each concentration NaCl and MgCl₂ are provided in **Supplementary Fig. 7** with the binding energy of possible individual chemical carbon contributions. As can be gathered from the figure, no carbonate species are detected in the C 1s spectra which generally appears at binding energies higher than 288.5 eV.⁶ This is important especially for this type of experiment as competitive ion sorption by other ions in solution (e.g., carbonates from CO₂ speciation) could introduce error in the Donnan Potential measurement.

Supplementary Fig. 7 Representative C 1s core level spectra collected from CR-61 membranes equilibrated with various concentrations of **a**, NaCl and **b**, MgCl₂ solution. The binding energy is calibrated by adjusting the aliphatic C to 285.0 eV. Circles represent the raw experimental data. Shirley background is included to be a visual guide to aid in tracking the vertical dashed lines representing binding energies of possible individual chemical carbon contributions. All binding energies are obtained from ref 6.

The reviewer is concerned that the authors do not present or discuss counter-ion (or co-ion, if available) results in the manuscript. Using the data provided (for reviewers only), the reviewer created plots for Na 1s and Mg 1s BE shifts versus bulk concentration below. No shift is observed for Na 1s. The same is true for Mg 1s up through 100 mM, with an interesting drop in BE at 1000 mM. This should be presented and discussed in light of the clear shifts observed for immobile membrane peaks. The spectra should also be available for the reader, creating simple stacked plots for all concentrations, one for Na 1s, one for Mg 1s, and one for Cl 1s (if available). A question that arises for the reviewer is, did the authors observe more than one peak for counter-ions or coion? For example, are there bulk and membrane specific peaks observed above a certain salt concentration? If the answer is yes, then what BE is being reported in plots

below? If the answer is no, then suggestions as to why would be interesting. Moreover, the reviewer would like to see a quantitative assessment of Na 1s/S 1s area ratio vs bulk concentration, Mg 1s/S 1s area ratio vs bulk concentration, and Cl 1s/S 1s (if available) vs. bulk concentration in order to assess potentially interesting adsorption properties. Only counter-ion BE's were provided to reviewers, not areas. This simple analysis makes Figure 1b more than just a cartoon. XPS can actually measure quantitatively the species present, and will further validate the Donnan model being put forth. If the membrane peaks can be separated from bulk peaks, these peak area ratios of membrane species vs. bulk solution concentration should be Langmuirian in nature.

Author reply: We thank our reviewer for evaluating our paper carefully, we understand your concerns about not presenting the counter-ion results. The S 1s data is all that is needed to convey the goals of this manuscript, which is to reveal the ability to measure the Donnan potential, which is why we wanted to focus on the S 1s. We believe that the counter-ion data are a complement to the S 1s data, but provide information with respect to the ions local environment. Our later experiments with different counter-ions form the foundation of a future study (not shown here) that we posit this behavior is counter-ion specific. We look forward to going more in depth with this study in a future manuscript as it is beyond the scope of this manuscripts work.

*As we stated earlier, no Cl 1s peak was observed for electrolyte concentrations lower than 1M and we did not see more than one peak for counter-ions up to 1 M concentration. Considering the nature of ion exchange membranes, **where co-ion concentration inside the membrane is negligible while counter-ion concentration is dictated by the number of fixed charges** (for CR-61 = 3.2 M), we believe that detected counter-ion specific peaks were mainly from the membrane phase for the measurements at low salt concentration. (Predicted membrane counter-ion concentrations for CR-61 equilibrated with 0.001-1 M NaCl and MgCl₂ using the classical Donnan model can also be found in Supplementary Table 11.) Although we are not able to deconvolute the contribution of solution layer cations from the detected Mg 1s core level, we anticipate that the drop in overall Mg 1s BE at 1 M solution concentration could arise from the additional contribution from the liquid layer (bulk electrolyte) which was not detectable in lower concentrations. It should be noted that even at the 1 M concentration no additional peak or apparent broadening of existing peaks for the S 1s, Na 1s and Mg 1s are observed because of the small binding energy differences between these components.*

The proposed counter-ion sorption behavior is also supported by quantitative analysis and numerical simulations on Na 1s/S 1s and Mg 1s/S 1s area ratios which do not show any obvious trend or changes up at low concentration (especially in our range), with some increase in the ratio at high concentration due to the increased signal contributions from the bulk solution becoming detectable.

The relevant discussion is provided in Supplementary Note 5-Binding energy shifts of other core levels and Supplementary Note 6-Numerical simulation of the photoelectron intensity

In the Supplementary Information:

We would like to note that the counter-ion binding energies show a different behavior. Contrary to our observations in the immobile membrane peaks, no clear concentration dependence is observed in the counter-ion related binding energy of NaCl- and MgCl₂-equilibrated CR-61 (see **Supplementary Fig. 9 and 10**). This behavior may arise from the weak interaction with the membrane sulfonate charges and/or the strong hydration shell of counter-ions. We are

currently devoting time and effort to better understand this phenomenon, and it will be the focus of a future manuscript.

In addition to counter-ions, co-ion (Cl 1s) core level spectra were also collected during the experiments, but no peak was observed for electrolyte concentrations lower than 1 M. Considering the nature of ion exchange membranes, where co-ion concentration inside the membrane is very low compare to the counter-ion concentration, which is dictated by the number of fixed charges (i.e., for CR-61 = 3.2 M), we believe that observed counter-ion specific peaks were coming mainly from the ions inside the membrane at lower salt concentration. This sorption behavior is supported by quantitative analysis and numerical simulations on Na 1s/S 1s and Mg 1s/S 1s peak area ratios given in **Supplementary Note 6**. As predicted, peak area ratios of counterion to membrane related core levels do not show any obvious trend or change at lower concentration. On the other hand, at 0.3 M salt concentration, Na 1s/S 1s and Mg 1s/S 1s area ratios starts to increase due to the additional contribution from detection of counter-ions in the liquid layer.

Since the detected Mg 1s and Na 1s regions at 1M solution concentration are a convolution of peaks coming from ions in both the membrane and liquid phase, the binding energy values at 1 M external solution concentration are excluded from the plot given in **Supplementary Fig. 10**. Unfortunately, it was not possible to deconvolute the solution phase ions from the ions inside the membrane because of small binding energy/electrical potential differences between them, which were not enough to form any peak separation or broadening. However, the small drop in overall Mg 1s BE at 1 M solution concentration (**Supplementary Table 6**. and **Fig. 9b**) may arise from an additional contribution from the liquid layer which was not detectable at lower concentrations.

Supplementary Fig. 9. Representative **a**, Na 1s and **b**, Mg 1s core level spectra collected from CR-61 membranes equilibrated with various concentrations of NaCl and MgCl₂ solutions. The binding energy is calibrated using bulk liquid phase water (LPW). Circles represent the raw experimental data, and lines indicate the sum of fits. The binding energy value of fitted core level components for each individual analysis position are provided in **Supplementary Table 6**.

Supplementary Fig. 10. Comparison of experimentally measured binding energy shifts in counterion and membrane specific core levels as a function of external **a**, NaCl and **b**, MgCl₂ solution concentration.

Supplementary Note 6.

Numerical simulation of the photoelectron intensity

Since we established that the thicknesses of the liquid layer on the CR-61 ion exchange membrane with various salt concentrations are similar in **Supplementary Note 1**, the sorption of counter-ions in CR-61 membranes over a range of external solution concentrations can be quantified from the relative intensities of the counter-ion peaks (Na 1s or Mg 1s) and membrane related S 1s. The simulations are built adopting a layered structure, where the membrane is simply buried underneath a salt solution layer of thickness d film. Accordingly, the intensity of the Na 1s and Mg 1s peaks at various external salt concentrations is obtained by integrating over the exponential escape probability as follows:

$$I_{Mg} = S_{Mg} n_{Mg}^m \lambda_{memb} e^{-d/\lambda_{solution}} + S_{Mg} n_{Mg}^s \lambda_{solution} \left[1 - e^{-d/\lambda_{solution}} \right] \quad (S8)$$

$$I_{Na} = S_{Na} n_{Na}^m \lambda_{memb} e^{-d/\lambda_{solution}} + S_{Na} n_{Na}^s \lambda_{solution} \left[1 - e^{-d/\lambda_{solution}} \right] \quad (S9)$$

where n_{Mg}^m , n_{Na}^m and n_{Na}^s , n_{Mg}^s are the number density of counter-ions inside the membrane and in the solution phase, respectively. The number density of counter-ions inside the membrane is estimated from the concentration of fixed charges in CR-61. λ_{memb} and $\lambda_{solution}$ are the IMFPs in the membrane and solution, respectively. The sensitivity constants, S_{Mg} and S_{Na} , are instrumental parameters which depend on the photoionization cross-section of elements at a given X-Ray energy, X-ray flux at a given X-ray energy, the orbital specific asymmetry, and the spectrometer efficiency for a given kinetic energy (KE). S must be taken into an account when quantifying photoelectron peaks with different KE.

Similarly, the intensity of the S 1s peaks of a CR-61 membrane underneath a liquid layer of thickness d , over the entire concentration range probed, is obtained from:

$$I_S = S_S n_S^m \lambda_{memb} e^{-d/\lambda_{solution}} \quad (S10)$$

By substituting the values of IMFPs listed in **Supplementary Table 9**, the relative intensities of the Na 1s or Mg 1s and S 1s XPS peaks are simulated over the entire salt concentration range probed. (**Supplementary Fig. 11**) The simulated intensity ratios agree with experimental observations. As expected, peak area ratios of counterion to membrane related core levels do not show any obvious trend or change at lower concentration. On the other hand, Na 1s/S 1s and Mg 1s/S 1s area ratios start to increase at high concentrations due to additional contribution from detection of counter-ions in the liquid layer. It needs to be highlighted that these intensity simulations were performed using homogeneous and well-defined layered structures and interfaces as an approximation of the real configuration, where most likely concentration gradients and mixed regions exist. In addition, it was previously established that the surface composition of the salt solutions are enhanced in the halide anion concentration (and thus attenuation of the cation) compared with the bulk of the solution at the liquid/vapor interface.^{14,15} Given these assumptions there is reasonable agreement between our experimental and simulated data, which reveal a similar trend of the primary detectable spectra contributions.

Supplementary Table 9. Estimated IMFPs for a given core level in the aqueous salt solution and the membrane using the Tanuma-Powell-Penn (TPP-2M) algorithm and the modified Bethe equation^{3,4}.

Core level	λ_{memb}	$\lambda_{solution}$
Na 1s	7.2 nm	9.2 nm
Mg 1s	6.7 nm	8.6 nm
S 1s	4.2 nm	5.3 nm

Supplementary Fig. 11. Experimental trends and simulated ratio of **a**, Na 1s and **b**, Mg 1s peak areas to S 1s as a function of external salt solution concentration. Purple bars represents the simulated intensity contribution of membrane while the blue bars represent the contribution from the solution layer. Error bars shows the experimental uncertainty, determined from the standard deviation of repeated measurements of four or more different positions in each case.

The revised discussion in quotes above makes no mention of the counter ions. The authors do not give a valid justification for this, and should include results and discussion of counter ion data as discussed by the reviewer above. Under revised Supplementary Note 5, Na 1s is mentioned at the end of the following sentence: “Supplementary Table 6 shows the binding energies of fitted XP spectra for O 1s-LPW, O 1s-GPW, O 1s-SO₃, S 1s-SO₃ and Na 1s.” The reviewer assumes this is a typo given the authors did not provide Na 1s data in Supplementary Table 6.

Author reply: We thank the reviewer for carefully reviewing of our manuscript and catching the above mentioned typo in Supplementary Information. According to the suggestion of the reviewer, we revised our Supplementary Information to present the B.E. positions of counter-ion related core levels and expanded the discussion of shifts in the other core levels as presented

above. Supplementary Table 6, presented below, is the revised version which now includes the binding energies of counter-ions.

In the Supplementary File:

Supplementary Table 6. Binding energy results of fitted core level peaks of membrane/solution system collected from four or five different positions for each electrolyte concentration

Name	P #1 B.E. (eV)	P #2 B.E. (eV)	P #3 B.E. (eV)	P #4 B.E. (eV)	P #5 B.E. (eV)
CR61- 1 mM NaCl					
O 1s-LPW	533.11	533.11	533.11	533.11	533.11
O 1s-GPW	535.28	535.30	535.25	535.28	535.33
O 1s- SO₃	531.59	531.62	531.60	531.58	531.60
S 1s-SO₃	2476.95	2476.96	2476.95	2476.92	2476.93
Na 1s	1071.62	1071.64	1071.63	1071.62	1071.64
CR61- 10 mM NaCl					
O 1s-LPW	533.11	533.11	533.11	533.11	533.11
O 1s-GPW	535.30	535.26	535.31	535.30	535.35
O 1s- SO₃	531.64	531.65	531.66	531.66	531.63
S 1s-SO₃	2477.00	2476.99	2476.98	2476.98	2476.97
Na 1s	1071.62	1071.62	1071.62	1071.66	1071.62
CR61- 30mM NaCl					
O 1s-LPW	533.11	533.11	533.11	533.11	533.11

O 1s-GPW	535.22	535.30	535.25	535.24	535.29
O 1s- SO₃	531.65	531.66	531.69	531.68	531.66
S 1s-SO₃	2476.98	2477.01	2477.01	2477.03	2477.00
Na 1s	1071.62	1071.64	1071.64	1071.64	1071.62
CR61- 0.1 M NaCl					
O 1s-LPW	533.11	533.11	533.11	533.11	533.11
O 1s-GPW	535.37	535.34	535.32	535.31	535.33
O 1s- SO₃	531.70	531.65	531.67	531.68	531.64
S 1s-SO₃	2477.00	2477.01	2477.00	2477.04	2477.03
Na 1s	1071.61	1071.60	1071.62	1071.63	1071.62
CR61- 0.3 M NaCl					
O 1s-LPW	533.11	533.11	533.11	533.11	533.11
O 1s-GPW	535.26	535.30	535.35	535.35	535.23
O 1s- SO₃	531.72	531.67	531.67	531.69	531.69
S 1s-SO₃	2477.04	2477.03	2477.02	2477.02	2477.04
Na 1s	1071.68	1071.68	1071.64	1071.62	1071.65
CR61- 1 M NaCl					

O 1s-LPW	533.11	533.11	533.11	533.11	533.11
O 1s-GPW	535.30	535.33	535.30	535.34	535.29
O 1s- SO₃	531.70	531.70	531.71	531.68	531.73
S 1s-SO₃	2477.07	2477.07	2477.05	2477.03	2477.06
Na 1s	1071.61	1071.68	1071.63	1071.63	1071.61
CR61- 1 mM MgCl₂					
O 1s-LPW	533.11	533.11	533.11	533.11	N/A
O 1s-GPW	535.26	535.15	535.21	535.24	N/A
O 1s- SO₃	531.60	531.60	531.63	531.62	N/A
S 1s-SO₃	2476.96	2476.96	2476.97	2476.97	N/A
Mg 1s	1304.53	1304.54	1304.54	1304.55	N/A
CR61- 10 mM MgCl₂					
O 1s-LPW	533.11	533.11	533.11	533.11	533.11
O 1s-GPW	535.23	535.19	535.20	535.24	535.18
O 1s- SO₃	531.64	531.65	531.63	531.61	531.63
S 1s-SO₃	2476.97	2476.99	2476.96	2476.99	2476.99
Mg 1s	1304.56	1304.52	1304.56	1304.53	1304.54
CR61- 30 mM MgCl₂					

O 1s-LPW	533.11	533.11	533.11	533.11	533.11
O 1s-GPW	535.25	535.28	535.19	535.22	535.23
O 1s- SO₃	531.67	531.67	531.64	531.67	531.63
S 1s-SO₃	2476.99	2477.00	2477.03	2476.98	2476.99
Mg 1s	1304.54	1304.53	1304.53	1304.57	1304.56
CR61- 0.1 M MgCl₂					
O 1s-LPW	533.11	533.11	533.11	533.11	533.11
O 1s-GPW	535.24	535.16	535.21	535.19	535.20
O 1s- SO₃	531.67	531.66	531.65	531.67	531.66
S 1s-SO₃	2477.01	2477.04	2477.02	2476.98	2477.01
Mg 1s	1304.56	1304.51	1304.50	1304.51	1304.54
CR61- 0.3 M MgCl₂					
O 1s-LPW	533.11	533.11	533.11	533.11	N/A
O 1s-GPW	535.14	535.11	535.16	535.17	N/A
O 1s- SO₃	531.65	531.67	531.65	531.66	N/A
S 1s-SO₃	2477.05	2477.03	2477.02	2476.99	N/A
Mg 1s	1304.52	1304.56	1304.59	1304.57	N/A

CR61- 1 M MgCl ₂					
O 1s-LPW	533.11	533.11	533.11	533.11	533.11
O 1s-GPW	535.14	535.19	535.05	535.01	535.09
O 1s- SO₃	531.68	531.68	531.70	531.69	531.68
S 1s-SO₃	2477.04	2477.04	2477.04	2477.06	2477.04
Mg 1s	1304.50	1304.50	1304.48	1304.46	N/A

These results are nice, showing no additional chemistry beyond what is expected. Are these results the same for all concentrations and experimental conditions? If so, an explicit statement should be made in the caption of Fig. S1 that says “No other chemistry was observed in the system *for all experiments*”. Should the S 1s label near ~230 eV be S 2s? Finally, if survey were collected out to Cl 1s (~2800 eV), this should also be shown.

Author reply:

We thank our reviewer for noticing our typo. Supplementary Fig. 1 and its caption have been revised as suggested.

In the Supplementary Information:

Supplementary Fig. 1. Survey spectra of membrane/solution/vapor system during the *in situ* tender-APXPS measurements for **a**, NaCl_(sol) equilibrated CR-61 and **b**, MgCl₂(_{sol}) equilibrated CR-61 membrane. No other chemistry was observed in the system for **all experiments**

It is unclear why Figure S6 is repeated in Figure S10.

Author reply:

We are sorry that Figure S6 and Figure S10 were unclear to our reviewer. Figure S6 shows the O 1s and S 1s core level spectra collected from CR-61 equilibrated with NaCl solutions while Figure S10 (Figure S13 in the latest version) shows the set of spectra collected for MgCl₂ equilibrated membranes.

Reviewer: 3

Comments:

The authors have adequately addressed my concerns.

Author reply: We thank our reviewer for evaluating our manuscript and finding it publishable in Nature Communications.

REVIEWERS' COMMENTS

Reviewer #2 (Remarks to the Author):

I commend the authors for bringing a fuller photoemission analysis to the reader. The authors have adequately addressed my concerns.

The Donnan Potential Revealed

Pinar Aydogan Gokturk¹, Rahul Sujanani², Jin Qian^{1,3}, Ye Wang^{1,3}, Lynn E. Katz⁴, Benny D. Freeman² and Ethan J. Crumlin^{1,3*}

¹ Advanced Light Source, Lawrence Berkeley National Laboratory, Berkeley, CA 94720, United States

² McKetta Department of Chemical Engineering, The University of Texas at Austin, Austin, Texas 78712, United States

³ Chemical Sciences Division, Lawrence Berkeley National Laboratory, Berkeley, California 94720, United States

⁴ Department of Civil, Architectural, and Environmental Engineering, The University of Texas at Austin, Austin, Texas 78712, United States

Dear Editor and Reviewers,

Thank you for the careful evaluations of our manuscript and providing the detailed comments, questions and suggestions. All the comments are very valuable and have been helpful for improving our manuscript.

We have addressed each comment very carefully, and have revised our manuscript accordingly. The original comments, questions and queries, and our responses are in *italics and in blue color* can be found below

Reviewer(s)' Comments to Author:

Reviewer: 2

Comments:

I commend the authors for bringing a fuller photoemission analysis to the reader. The authors have adequately addressed my concerns.

Author reply: We thank our reviewer for evaluating our manuscript and finding it publishable in Nature Communications.